# Focused Diffusion-GAN: Object-Centric Image Generation Using Integrated GAN and Diffusion Frameworks

## Abstract

Generative Adversarial Networks (GANs) and Diffusion Models (DMs) have shown significant progress in synthesizing high-quality object-centric images. However, generating realistic object-centric images remains challenging when training datasets are limited or contain degraded images (e.g., privacy-induced face blurring). Under these conditions, existing generative models frequently produce images that lack perceptual quality or exhibit overfitting to the training examples. To overcome these limitations, we propose a novel hybrid generative model, *Focused Diffusion-GAN (FDGAN)*, targeting low-data object-centric regimes, which integrates a GAN discriminator directly into the diffusion model at intermediate denoising stages. Central to FDGAN is an Additional Noise Perturbation Module (ANPM) that selectively activates the GAN component only for images sufficiently denoised, ensuring the discriminator receives meaningful input. Additionally, ANPM applies targeted noise perturbations within predefined bounding-box regions, implicitly guiding the model's focus toward key objects. FDGAN differs from other models like LayoutDiffusion, which explicitly conditions synthesis on fixed bounding-box layouts, or Diffusion-GAN and StyleGAN2-ADA, which employ noise augmentation throughout the entire training process, by combining adversarial training with targeted noise perturbations at specific intermediate diffusion steps. We evaluate FDGAN on three small object-centric datasets (Cityscapes subset, Traffic-Signs, and MS-COCO "potted plant") and, against strong GAN, diffusion, and object-centric baselines, show improved perceptual quality (Fréchet Distance) and reduced overfitting (Feature Likelihood Score). Ablation studies indicate that selective mid-timestep adversarial guidance together with ANPM improves the realism–overfitting trade-off in limited-data generative tasks.

## 1 Introduction

Object-centric image generation has gained significant attention due to its ability to produce high-quality images where specific objects are accurately and realistically represented (Nichol & Dhariwal, 2021; Wang et al., 2023). This capability is crucial in various domains. For example, in smart-city applications, object-centric generation supports realistic simulations of urban environments for traffic analysis and emergency response planning (Mohammadi & Al-Fuqaha, 2018). In manufacturing, generating diverse images of rare defects helps train accurate defect-classification models when actual defective samples are scarce (Zhong et al., 2023; Duan et al., 2023). Various generative models have rapidly evolved to meet these needs, including Variational Autoencoders (VAEs) (Kingma & Welling, 2022; Doersch, 2021), Energy-Based Models (EBMs) (Lee et al., 2023; Yang & Ji, 2023; Yu et al., 2023), GANs (Goodfellow et al., 2020; Salimans et al., 2016), and Diffusion Models (DMs) (Ho et al., 2020; Nichol & Dhariwal, 2021).

Among these, GANs and DMs have emerged as leading methodologies due to their superior performance in generating high-quality images (Chakraborty et al., 2023). GANs have demonstrated remarkable realism, whereas DMs are known for stable training and modeling complex distributions (Nichol & Dhariwal, 2021). However, both methods individually struggle under limited-data conditions, motivating hybrid approaches that leverage their complementary strengths.

Despite significant advances, generating high-quality object-centric images remains challenging, especially under severely limited or degraded training data conditions (e.g., privacy-induced face blurring) (Karras et al., 2020a; Sauer et al., 2021; Wang et al., 2023; Zhao et al., 2020; Noguchi & Harada, 2019). Under these constraints, models often show poor quality (Sauer et al., 2021; Noguchi & Harada, 2019), limited diversity (Karras et al., 2020a; Dubiński et al., 2023; Zhao et al., 2020; 2022), and overfitting (Karras et al., 2020a; Wang et al., 2023; Zhao et al., 2020; 2022). Small or degraded datasets often cause replication of training samples, limiting generalization for downstream tasks like object detection (Ultralytics, 2022; Jocher et al., 2023; Jocher & Qiu, 2024). Ensuring robust generalization, perceptual fidelity, and diverse outputs under these conditions is thus a central challenge in generative modeling research (Karras et al., 2020a; Sauer et al., 2021; Zhao et al., 2020; Bau et al., 2019).

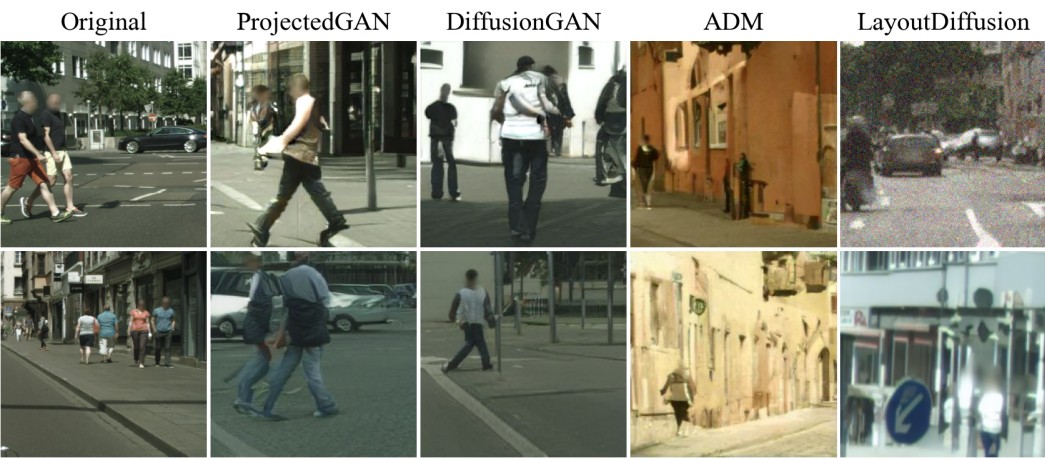

Figure 1: Artifacts in generative models under limited-data Cityscapes: structural inaccuracies, unrealistic textures, blurred details.

Fig. 1 presents examples generated by four recent generative models—Projected GAN, Diffusion-GAN, ADM, and LayoutDiffusion—to illustrate common issues in object-centric image generation under constrained conditions. Projected GAN and Diffusion-GAN frequently produce distorted or anatomically inconsistent objects. ADM often fails to preserve original domain characteristics, resulting in ambiguous or blurred images. LayoutDiffusion, despite explicitly conditioning on layouts, sometimes generates outputs lacking critical details. The first column provides original dataset images for visual reference. Recent diffusion-based methods like LayoutDiffusion (Zheng et al., 2024), which explicitly condition on bounding-box (BB) layouts, improve compositional coherence but typically require large datasets to prevent overfitting.

Addressing these limitations, we propose *Focused Diffusion-GAN (FDGAN)*, a hybrid generative model that integrates a GAN discriminator $D$ directly into a diffusion-based generator. FDGAN leverages adversarial feedback *selectively at intermediate diffusion steps* and introduces an *Additional Noise Perturbation Module (ANPM)* that applies BB-localized perturbations so that $D$ sees meaningful partially denoised inputs. This early-timestep adversarial window and localized perturbation act as an implicit attention signal, improving realism and object fidelity in low-data regimes while leaving inference cost unchanged.

*Scope and intended use-case.* FDGAN is explicitly designed for *limited-annotation, object-centric* regimes—on the order of ∼2–3k labeled crops per class, often with degraded content (e.g., privacy blur or occlusion). Such settings are common in practice, where large-scale annotation is infeasible or inefficient. FDGAN therefore targets scenarios that require effective generation under scarce supervision, prioritizing generalization, reduced overfitting, and perceptual fidelity. This design choice directly informs our dataset selection and evaluation protocol. In summary, FDGAN is a low-data, object-aware synthesizer for augmenting downstream detectors (e.g., YOLO/DETR), rather than a general, large-scale text-to-image model.

Our key contributions are:

- We propose FDGAN, a hybrid generative model that integrates adversarial training into the *intermediate* stages of a diffusion-based generator, enhancing the quality and diversity of object-centric images under *limited or degraded* data conditions (see Section 3).

- We introduce the *Additional Noise Perturbation Module (ANPM)*, which selectively activates adversarial training only when samples are sufficiently denoised and injects targeted noise within object-specific BB regions, implicitly guiding attention toward critical object details (see Section 3.2).

- We devise a *selective early-timestep adversarial window* together with a *timestep-aware loss aggregation* schedule $\lambda_{\mathrm{GAN}}(t)$, enabling stable training in low-data settings with *no inference-time overhead*; ablations show each component is necessary (see Section 3.2 and Appendix F).

Furthermore, we evaluate FDGAN under constrained dataset conditions and benchmark it against diffusion-only (e.g., LayoutDiffusion), GAN-only (e.g., Projected GAN, StyleGAN2-ADA), and hybrid (e.g., Diffusion-GAN) approaches. *Our evaluation spans three curated object-centric datasets: a Cityscapes–Pedestrian subset with privacy-blurred faces, Traffic-Signs, and an MS-COCO "potted plant" subset.*

## 2 RELATED WORK

Object-centric image generation focuses on producing high-quality, detailed images in which specific class objects are accurately represented (Nichol & Dhariwal, 2021; Wang et al., 2023; Sauer et al., 2021; Jiang et al., 2023). This task remains challenging due to the high dimensionality and structural complexity inherent in real-world images, particularly when available training data is limited or hard to obtain.

Likelihood-based methods, such as Variational Autoencoders (VAEs) (Kingma & Welling, 2022; Kingma & Dhariwal, 2018), offer stable optimization and high-resolution synthesis, but typically sacrifice perceptual realism compared to GANs. Object-centric VAEs like Multi-Object VAE (MONet) (Burgess et al., 2019) attempt to decompose scenes into individual objects; however, their performance often declines with increased scene complexity, occlusion, or interactions between objects. Similarly, Energy-Based Models (EBMs), including object-centric EBMs (OC-EBMs) (Zhang et al., 2022), can model object interactions effectively but frequently struggle to capture fine details and accurate object boundaries, leading to fragmented or merged object representations.

Diffusion Models (DMs) have recently emerged as powerful generative frameworks, synthesizing images by reversing a gradual noise addition process. UNet-based DDPMs and ADM (Ho et al., 2020; Nichol & Dhariwal, 2021) and transformer-based DiT-XL/2-G (Peebles & Xie, 2023) significantly advanced image quality, although often at substantial computational cost. Recent diffusion-based methods increasingly leverage explicit object-centric representations to enhance spatial coherence. For example, Object-Centric Slot Diffusion (Wu et al., 2023) conditions latent diffusion on object slots to better maintain object integrity. Additionally, LayoutDiffusion (Zheng et al., 2024) directly conditions the generation of BB layouts, effectively improving compositional coherence and explicit spatial control. However, these purely diffusion-based models typically require large datasets to generalize effectively, limiting practical applicability when only small or specialized datasets are available.

Generative Adversarial Networks (GANs) (Goodfellow et al., 2014) excel at generating high-resolution, perceptually realistic images (Brock et al., 2018; Karras et al., 2020b) but often suffer from unstable optimization and incomplete data coverage (Arjovsky et al., 2017; Heusel et al., 2017; Mescheder et al., 2018; Metz et al., 2017). Models such as Projected GAN (Sauer et al., 2021) and Pedestrian-Synthesis GAN (PSGAN) (Ouyang et al., 2018) specifically enhance GAN stability and object-centric generation, yet their effectiveness is highly domain-specific and sensitive to model configuration. StyleGAN2-ADA (Karras et al., 2020a) addresses training instability through adaptive $D$ augmentation, and Pix2PixHD (Isola et al., 2018; Wang et al., 2018a) uses conditional adversarial networks for spatial coherence; however, these models are constrained by the need for extensive training pairs and remain limited in generalizability across diverse object classes. Diffusion-GAN (Wang et al., 2023) integrates diffusion-based noise models into GANs to enhance stability, but the approach remains computationally expensive and challenging to implement.

Prior hybrids that mix adversarial objectives with diffusion have shown promise—including GAN-centric variants such as Diffusion-GAN (Wang et al., 2023) and more recent diffusion-centric (distillation-style) hybrids. Our setting is complementary: FDGAN targets *limited-annotation, object-centric* regimes and couples three elements that, to our knowledge, have not been jointly explored for this use case: (i) a *selective early-timestep* adversarial window so the $D$ only sees partially denoised, informative states; (ii) *bounding-box–localized* perturbations (ANPM) as a lightweight spatial cue instead of explicit layout conditioning; and (iii) a *timestep-aware* adversarial weighting schedule. This combination targets small-data stability and spatial fidelity without inference-time cost. Removing any single component degrades DINOv2 metrics and increases overfitting (Table 2).

## 3    FDGAN: METHOD AND THEORETICAL ANALYSIS

FDGAN synthesizes high-quality object-centric images by integrating a PatchGAN-style $D$ (Isola et al., 2017; Wang et al., 2018a) into the DM derived from ADM (Nichol & Dhariwal, 2021). Unlike existing diffusion models (e.g., LayoutDiffusion (Zheng et al., 2024), GLIGEN (Li et al., 2023)), that rely solely on conditioning signals (attention or embeddings) to guide object generation, FDGAN incorporates adversarial feedback at intermediate diffusion steps. Central to FDGAN is our proposed *Additional Noise Perturbation Module (ANPM)*, which selectively activates the GAN component at specific intermediate timesteps, ensuring the $D$ receives partially denoised image pairs. Additionally, the ANPM injects targeted Gaussian noise within predefined BB regions, implicitly guiding the model's attention to object details, thus improving spatial coherence, realism, and generative diversity.

### 3.1    BACKGROUND FOR FDGAN

FDGAN's GAN component operates following the principles described by (Goodfellow et al., 2014), based on a competitive framework between two neural entities: a $G$ and a $D$. The $G$ typically converts random vectors into synthetic data instances, while the $D$ evaluates these synthetic outputs alongside real data samples, classifying each as either genuine or artificial. This dynamic results in a min-max game described by the adversarial objective function (Goodfellow et al., 2020):

$$\min_{\mathbf{G}} \max_{\mathbf{D}} V(G, D) = \mathbb{E}_{x \sim p_{dt}}[\log D(x)] + \mathbb{E}_{z \sim p_x}[\log(1 - D(G(z)))], \tag{1}$$

where $x \sim p_{dt}$ represents samples from the real data distribution, and $z \sim p_x$ denotes samples from the noise prior. In FDGAN, $z$ corresponds specifically to *partially denoised* diffusion states rather than purely random vectors. Unlike conventional GANs, where $G$ starts from unstructured noise, FDGAN leverages these intermediate states as more informative adversarial inputs. This allows $G$ to refine its outputs progressively across timesteps, yielding increasingly realistic generations.

The diffusion component follows the standard forward process (Song et al., 2022; Sohl-Dickstein et al., 2015):

$$q(x_{1:T}|x_0) = \prod_{t=1}^{T} q(x_t|x_{t-1}), \quad q(x_t|x_{t-1}) = \mathcal{N}(x_t; \sqrt{1-\beta_t}\, x_{t-1}, \beta_t \mathbf{I}), \tag{2}$$

with variance schedule $\beta_t$. Equivalently, a noisy latent at timestep $t$ can be sampled directly from clean data $x_0$ as

$$q(x_t|x_0) = \mathcal{N}\left(x_t; \sqrt{\bar{\alpha}_t}x_0, (1 - \bar{\alpha}_t)\mathbf{I}\right), \tag{3}$$

where $\bar{\alpha}_t = \prod_{i=1}^{t}(1 - \beta_i)$. This formulation enables efficient sampling of intermediate states and stable training of the reverse denoising process.

The UNet in FDGAN serves as both the neural network for the computation of diffusion loss and the $G$ in the GAN framework. For timesteps $t < t_{\text{early}}$, the ANPM selectively modifies the image content within predefined BB regions by adding Gaussian noise consistent with the diffusion noise level at that specific timestep. These modified images are then input to the UNet, implicitly guiding the model's attention toward these targeted BB regions, enhancing object coherence and spatial fidelity in the generated images. A detailed description of the UNet's architecture is provided in the FDGAN Architecture section.

Our practical motivation for FDGAN stems from scenarios where datasets are small or costly to collect, yet precise object-focused generation is essential. Accordingly, we evaluate on three representative

settings—Cityscapes–Pedestrian (occluded urban scenes), Traffic-Signs (simpler structured objects), and COCO potted plants (diverse objects/backgrounds)—to highlight robustness across complexity. We also employ a modified sampling strategy in which the model processes fully noised inputs and receives BB-localized perturbations as conditional cues (details in Appendix B).

By incorporating these steps, FDGAN improves the generative process, producing high-quality, contextually accurate images. The architecture leverages diffusion's stability and coverage while mitigating GAN training instability and limited diversity.

## 3.2 FDGAN ARCHITECTURE

The overview of FDGAN is presented in Fig. 2. The images follow the original diffusion model diffusion paths, forward diffusion (green blocks $1 \rightarrow 3$) and reverse (green blocks $4 \rightarrow 6$) shown in the figure. During reverse diffusion (blocks 4–6), the U-Net predicts the noise and computes $L_{\text{diffusion}}$. In FDGAN we introduce three new modules (orange blocks): *ANPM* (7), the *GAN branch* comprising the denoised–image output and the $D$ (8→10), and the *Loss Aggregator* (11). These additions supply the GAN and $L_{recon}$ losses, which are then combined with the diffusion loss to update the $G$.

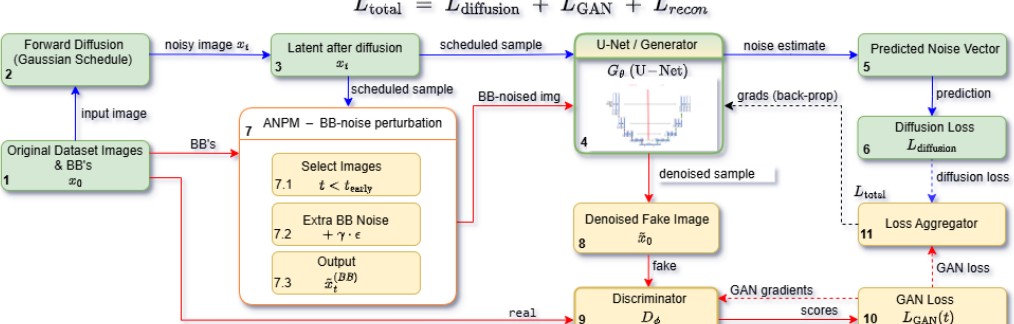

Figure 2: **FDGAN architecture overview.** Green indicates original diffusion components; orange indicates proposed GAN components (ANPM, $D$, Loss Aggregator). Solid arrows indicate data flow; dashed arrows indicate gradient flow.

The ANPM plays two critical roles in FDGAN: selectively activating adversarial training at intermediate diffusion steps controlled by the hyperparameter $t_{\text{early}}$, and applying targeted noise perturbations within BB regions, guiding the model's attention toward important object details. These targeted perturbations help enhance the model's focus, realism, and object-level coherence. FDGAN synthesizes object-centric images via four integrated steps: (1) an *Add-BB-noise Step* (block 7), applying targeted perturbations within BB regions at specific diffusion timesteps; (2) a *Sampling Step* (blocks 4, 8), denoising these perturbed samples via the U-Net $G$ to produce partially denoised images and estimate diffusion losses; (3) a *GAN Step* (blocks 8–10), where the $D$ evaluates generated images against real ones, computing adversarial losses; and (4) a *Loss Aggregation Step* (block 11), combining diffusion, GAN, and reconstruction losses and backpropagating gradients to update $G$ and $D$.

Specifically, the *Add-BB-noise Step* (block 7—ANPM) explicitly enhances object-level coherence and realism. At timesteps satisfying $t < t_{\text{early}}$, the ANPM processes selected samples sequentially: first, the *Select Images* sub-module (block 7.1) identifies partially denoised samples; next, the *Extra Noise* sub-module (block 7.2) injects targeted Gaussian noise ($+\gamma\epsilon$) selectively within BB regions encapsulating objects of interest (see Algorithm 1 in Appendix D); finally, the *Output* sub-module (block 7.3) forwards these BB-noised samples ($\tilde{x}_t^{(BB)}$) to the U-Net generator $G$. Subsequently, $G$ denoises these perturbed inputs to produce partially denoised fake images ($\tilde{x}_0$) and simultaneously estimates the noise ($\hat{\epsilon}_\theta(x_t, t)$) necessary for computing $L_{\text{diffusion}}$ (blocks 4–6). The discriminator $D_\phi$ then evaluates the realism of these partially denoised images against corresponding real samples ($x_0$), producing the GAN loss $L_{\text{GAN}}(t)$. Finally, diffusion, GAN, and reconstruction losses are aggregated

(block 11) according to the timestep-dependent weighting strategy detailed in Appendix F, updating $G$ parameters $\theta$, while $D$ parameters $\phi$ are updated solely via $L_{\text{GAN}}$.

**Generator/UNet.** To ensure our model's performance is not tied to a specific architecture, we utilize a UNet design with attention mechanisms and timestep embeddings, known to be highly effective in image-processing tasks. Our UNet comprises multiple down-sampling and up-sampling levels: specifically, 4 layers each for 64x64 inputs, and 5 layers each for 128x128 inputs. Each level integrates residual blocks (ResBlocks) as described in (He et al., 2016), and attention blocks to enhance feature representation, similar to recent approaches (Park et al., 2019; Sun & Wu, 2019; Wang et al., 2018a). The $G$ receives both the image and timestep embeddings, conditioning the batch normalization parameters within the UNet on timesteps to adapt dynamically to varying noise levels during training. Furthermore, $G$ employs a channel multiplier at each level, attention at specified resolutions, and optional convolutional up-sampling and down-sampling, ensuring flexibility and robust high-quality image generation.

**Discriminator.** The $D$ in FDGAN assesses the realism of generated images relative to real images. We incorporate a three-layer *PatchGAN D* (N-Layer, fully convolutional) similar to that used in Pix2PixHD (Isola et al., 2017; Wang et al., 2018a). This design captures local high-frequency details while keeping the parameter count modest. Only images from timesteps $t < t_{\text{early}}$, where $G$ inputs are already largely denoised, participate in adversarial training, ensuring meaningful image pairs for discrimination. For instance, in a 4,000-step diffusion chain, we set $t_{\text{early}} = 400$. The $D$ employs a least squares GAN (LSGAN) loss, stabilizing training and gradient behavior. The targeted noise applied within BB regions by the ANPM provides an implicit attention mechanism, guiding the $G$ toward enhanced object realism, finer details, and coherent object placement. Ablation studies confirm that disabling the GAN component (setting $t_{\text{early}} = 0$) notably reduces image quality and detail, validating the effectiveness of $D$ integration.

### 3.3 LOSS FUNCTIONS

The total objective combines diffusion, adversarial, and pixel-level reconstruction terms:

$$L_{\text{total}}(t) = L_{\text{diffusion}} + \lambda_{\text{GAN}}(t)\,L_{\text{GAN}}(t) + L_{\text{recon}}(t), \tag{4}$$

where $\lambda_{\text{GAN}}(t)$ is the dynamic weight provided by the scaling algorithm implemented in the *Loss Aggregator* (see Appendix F), and where $L_{\text{recon}}(t)$ represents reconstruction losses defined below.

**GAN loss** is defined as:

$$L_{\text{GAN}}(t) = \lambda_D(t)\,\mathbb{E}_{x\sim p(x),\,t<t_{\max}}\left[\log D(x)\right] + \lambda_G(t)\,\mathbb{E}_{z\sim p(z),\,t<t_{\max}}\left[\log\left(1 - D(G(z))\right)\right], \tag{5}$$

where $\lambda_D(t)$ and $\lambda_G(t)$ are timestep-dependent weights balancing $D$ and $G$ signals; $\mathbb{E}[\cdot]$ denotes expectation over the data distribution $p(x)$ or latent noise distribution $p(z)$. The term $\mathbb{E}_{x\sim p(x),t<t_{\max}}[\log D(x)]$ calculates the expected log-likelihood of the $D$ correctly identifying real images, while $\mathbb{E}_{z\sim p(z),t<t_{\max}}[\log(1 - D(G(z)))]$ measures the expected log-likelihood of the $D$ incorrectly identifying generated images as real, thus optimizing the $G$'s ability to fool $D$.

**Diffusion loss:** aims to reconstruct the original data from its noisy, diffused state by training the model to predict and remove noise at each timestep. Following the standard approach in diffusion modeling introduced by Nichol & Dhariwal Nichol & Dhariwal (2021), the diffusion loss is defined as the mean squared error between the true noise and the model's predicted noise:

$$L_{\text{diffusion}} = \mathbb{E}_{x_0\sim q(x_0),\,t\sim\mathcal{U}(1,T),\,\epsilon\sim\mathcal{N}(0,I)}\left[\|\epsilon - \epsilon_\theta(x_t,t)\|^2\right], \tag{6}$$

where $x_t = \sqrt{\bar{\alpha}_t}\,x_0 + \sqrt{1 - \bar{\alpha}_t}\,\epsilon$, and $\epsilon \sim \mathcal{N}(0, I)$ is the Gaussian noise added to the clean image $x_0$. Here, $\epsilon_\theta(x_t, t)$ represents the model's prediction of the noise component at timestep $t$. In practice, at each training step, we sample a clean image $x_0$ from the real data distribution $q(x_0)$ and a timestep $t$ uniformly from the total timesteps $T$. We then generate a noisy image $x_t$ by adding Gaussian noise scaled by the timestep-specific schedule $\bar{\alpha}_t$. The neural network (a UNet in our case) is tasked with estimating the noise $\epsilon$. Minimizing this loss allows the model to progressively predict and remove noise at each diffusion step, effectively reversing the diffusion and reconstructing high-quality images from noisy inputs.

**Reconstruction losses:** to stabilize training and encourage pixel-level fidelity, we explicitly define the reconstruction loss as a combination of global and targeted components:

$$L_{\text{recon}}(t) = \lambda_{L1}(t)L_1 + \lambda_{\text{enhanced}}L_{\text{enhanced}}, \tag{7}$$

where

$$L_1 = \mathbb{E}_{x,\hat{x}}\big[\|x - \hat{x}\|_1\big], \qquad L_{\text{enhanced}} = \mathbb{E}\big[\big((G(z) - x)^2 \cdot M_{\text{BB}}\big)\big]. \tag{8}$$

Here, $\lambda_{L1}(t)$ dynamically adjusts the contribution of global $L_1$ loss across different timesteps $t$, while $\lambda_{\text{enhanced}}$ is a fixed hyperparameter emphasizing detailed reconstruction within BB regions via mask $M_{\text{BB}}$. By combining Eqs. equation 5, equation 6 and 7 inside Eq. equation 4, FDGAN balances diffusion-based denoising with adversarial realism and pixel-level accuracy, yielding improved object-centric synthesis. This careful balancing of objectives is crucial for stable training and helps FDGAN produce high-quality, diverse images even from limited or degraded datasets.

### 3.4 Implementation Details

Our FDGAN model [1], implemented in PyTorch (Paszke et al., 2019), was trained on two A100 GPUs using staged training: diffusion-only followed by adversarial fine-tuning. We employ a UNet $G$ with GroupNorm32 (Nichol & Dhariwal, 2021) and an N-Layer PatchGAN $D$. Full architectural and training details are provided in Appendix E.

**Pretrained initialization and early stopping.** For the COCO *potted plant* experiments we fine-tune author-released checkpoints rather than training from scratch, following evidence that transfer from a strong source generator is beneficial in low-data regimes (faster convergence, higher quality) (Wang et al., 2018b; Grigoryev et al., 2022). During fine-tuning we generate validation samples every 20k steps and select the snapshot that minimizes DINOv2 FD while maintaining Recall; training is stopped once these metrics plateau or begin to degrade, consistent with metric-driven early stopping in generative modeling (e.g., FID/precision–recall and, when available, diversity proxies such as MS-SSIM) (Heusel et al., 2018; Kynkäänniemi et al., 2019; Wang et al., 2004). A detailed, reproducible protocol (including unconditional vs. conditional cases and qualitative checks) is provided in Appendix J (COCO-specific training utilities in Appendix E.2).

## 4 Evaluation of FDGAN

For FDGAN's evaluation, we curated three small object-centric datasets (each $< 3k$ images): a Cityscapes–Pedestrian subset, a Traffic-Signs dataset, and an MS-COCO "potted plant" subset. All datasets were standardized to $256 \times 256$ crops with bounding-box guidance. This setup spans challenging occlusion-heavy urban scenes, fine-detail signage, and semantically diverse object scenes with complex backgrounds (COCO), all under low-data constraints (details in Appendix C).

**Comparative evaluation.** We benchmark FDGAN against three families of baselines: (1) *GAN-only* (Projected GAN (Sauer et al., 2021), StyleGAN2-ADA (Karras et al., 2020a), PSGAN (Ouyang et al., 2018), Pix2Pix (Isola et al., 2018), and our proposed OC-ProjectedGAN (an object-centric adaptation of Projected GAN detailed in Appendix I); (2) *Diffusion-only* (ADM (Nichol & Dhariwal, 2021), DiT-XL/2-G (Peebles & Xie, 2023), LayoutDiffusion (Zheng et al., 2024)); and (3) the *hybrid* Diffusion-GAN (Wang et al., 2023), integrating diffusion-based noise into GAN training. All explicitly object-centric baselines (PSGAN, Pix2Pix, LayoutDiffusion, Diffusion-GAN, OC-ProjectedGAN) are trained with the same BB masks as FDGAN for a fair comparison.

We evaluate models using two complementary sets of metrics computed via the official implementation [2]: (1) DINOv2-based metrics (Stein et al., 2023), including Fréchet Distance (FD), FD$\infty$, Kernel Distance (KD), Feature Likelihood Score (FLS), FLS overfit, and Coverage and Fidelity (CT, CT mod); and (2) traditional Inception-V3 metrics (Kynkäänniemi et al., 2019; Meehan et al., 2020), including FID, Precision, Recall, Density, and Coverage. Our primary analysis emphasizes DINOv2 metrics due to their stronger alignment with perceptual quality in object-centric scenarios (detailed definitions provided in Appendix G).

### 4.1 Comparison with Conventional GAN and Diffusion Models

To benchmark FDGAN, we compare its performance against recent conventional GAN and DM models. Specifically, we select ADM (Nichol & Dhariwal, 2021), DiT-XL/2-G (Peebles & Xie,

---

[1] Code and data will be released upon acceptance

[2] https://github.com/layer6ai-labs/dgm-eval

2023), Projected GAN (Sauer et al., 2021), and StyleGAN2-ADA (Karras et al., 2020a), along with Diffusion-GAN (Wang et al., 2023), a hybrid approach that integrates diffusion into GAN training for better sample diversity. Explicit object-centric models are analyzed separately in Section 4.2. *Table 1* summarizes the quantitative results for all three datasets. On the Cityscapes–Pedestrian subset,

Table 1: Results on the three datasets using DINOv2 (FD, KD, FLS Overfit) and Inception-V3 (FID), where * denotes explicitly object-centric models or our modifications. Lower is better for FD, KD, and FID; FLS Overfit closer to 0 indicates less overfitting.

| Model | Cityscapes–Pedestrian | | | | Traffic-Signs | | | | COCO (Potted Plant) | | | |
|---|---|---|---|---|---|---|---|---|---|---|---|---|
| | FD↓ | KD↓ | FLS-O | FID↓ | FD↓ | KD↓ | FLS-O | FID↓ | FD↓ | KD↓ | FLS-O | FID↓ |
| PSGAN* | 774.13 | 5.01 | −25.33 | 78.56 | – | – | – | – | – | – | – | – |
| Projected GAN | 828.32 | 3.98 | −11.93 | 15.81 | 586.91 | 2.24 | −42.60 | 18.18 | 1094.78 | 1.78 | −32.86 | 44.15 |
| OC-ProjectedGAN* | 1076.03 | 5.12 | −34.87 | 22.26 | 693.43 | 2.93 | −43.66 | 33.66 | 1316.79 | 2.52 | −33.06 | 61.75 |
| Pix2Pix* | 958.23 | 4.27 | −27.64 | 88.78 | 828.10 | 3.16 | −48.53 | 116.92 | 1623.55 | 3.98 | 42.73 | 62.79 |
| StyleGAN2-ADA | 1948.78 | 8.77 | −48.73 | 71.36 | 1487.25 | 5.86 | 37.57 | 96.84 | 1450.62 | 3.04 | −39.26 | 76.57 |
| ADM | 1275.56 | 4.89 | −19.35 | 42.61 | 551.36 | 2.18 | 33.20 | 34.76 | 935.42 | 1.19 | −31.86 | 56.68 |
| DiT-XL/2-G | 2254.54 | 11.64 | −49.13 | 92.65 | 1349.91 | 4.63 | −47.66 | 140.43 | 926.54 | 1.53 | −30.93 | 56.10 |
| **FDGAN*** | **583.70** | **2.83** | **1.39** | 19.16 | **416.19** | **1.54** | **−22.49** | 28.19 | **889.95** | **1.17** | **−28.93** | 43.83 |
| Diffusion-GAN | 920.67 | 4.33 | −19.33 | **14.80** | 616.85 | 2.40 | −45.63 | **16.88** | 1010.54 | 1.61 | −32.40 | **30.71** |
| LayoutDiffusion* | 1313.52 | 5.27 | −46.47 | 75.00 | 680.03 | 2.74 | −42.80 | 57.65 | 1047.62 | 1.62 | −33.93 | 62.95 |

FDGAN achieves the lowest DINOv2 FD of 583.70, improving by 32.6% over PSGAN (774.13) and by 41.9% compared to Projected GAN (828.32). It also surpasses ADM in Inception-V3 FID (19.16 vs. 42.61), demonstrating the benefit of adding adversarial feedback to diffusion. On the Traffic-Signs dataset, FDGAN again yields the lowest DINOv2 FD of 416.19, outperforming Projected GAN by 29.1% (586.91) and Diffusion-GAN by 32.5% (616.85), while maintaining competitive FID (28.19 vs. 16.88 for Diffusion-GAN). Finally, on the COCO *potted plant* subset, FDGAN achieves a DINOv2 FD of 901.76, improving by 17.6% compared to Projected GAN (1094.78) and by 9.7% relative to Diffusion-GAN (1010.54). FDGAN also improves KD and FLS Overfit over all baselines, indicating stronger distributional alignment and reduced overfitting even in this semantically diverse and background-rich setting. Although GAN-centric models sometimes achieve lower raw FID on simpler domains, FDGAN provides a better overall balance, combining realism, diversity, and generalization across all three datasets. A complete set of evaluation metrics—including Precision, Recall, Density, Coverage, FD∞, and CT/CT-mod—is provided in Appendix G, while qualitative comparisons appear in Appendix H.

*Reproducibility note.* COCO results were obtained with a training variant detailed in Appendix E.2 (same FDGAN core, minor training utilities—ROI-focused banding and a small ROI discriminator—enabled for this dataset).

### 4.2 COMPARISON WITH OBJECT-CENTRIC METHODS

To further assess FDGAN in explicitly object-centric settings, we benchmark it against specialized models designed for object-conditioned generation. These include LayoutDiffusion (Zheng et al., 2024) (evaluated under low-data conditions), OC-ProjectedGAN (our object-centric adaptation of Projected GAN), PSGAN (Ouyang et al., 2018) (only for Cityscapes), and Pix2Pix (Isola et al., 2018). FDGAN outperforms these baselines across all three datasets. On Cityscapes–Pedestrian, FDGAN achieves a DINOv2 FD of 583.70, improving by 55.6% over LayoutDiffusion (1313.52), 24.6% over PSGAN (774.13), and 39.1% over Pix2Pix (958.23). On Traffic-Signs, FDGAN's FD of 416.19 represents gains of 32% over LayoutDiffusion (680.03) and 34% over OC-ProjectedGAN (693.43). On the COCO *potted plant* subset, FDGAN obtains an FD of 901.76, improving by 14% compared to LayoutDiffusion (1047.62) and 31% relative to Pix2Pix (1623.55). These results confirm that integrating adversarial training selectively into diffusion, combined with BB-localized perturbations, yields stronger fidelity, detail, and diversity than models relying on explicit layouts or paired supervision.

Across datasets, FDGAN achieves quantitatively stronger results on the simpler Traffic-Signs set, as expected given its reduced complexity. Nevertheless, FDGAN also ranks highest across DINOv2-based metrics on the more challenging Cityscapes and COCO subsets, achieving the best overall trade-off between realism, diversity, and generalization. Specifically, FDGAN exhibits the lowest

Fréchet Distance (FD), improved Kernel Distance (KD), and FLS Overfit values closest to zero, indicating both higher realism and reduced memorization. Moreover, FDGAN maintains balanced coverage and fidelity in the extended DINOv2 metrics (CT and CT-mod), confirming robustness across dense urban scenes, fine-detail signage, and semantically diverse household objects. The full set of evaluation metrics is reported in Table 6 (Appendix G), with qualitative comparisons in Appendix H.

### 4.3 ABLATION STUDIES

We conducted ablations on the Cityscapes–Pedestrian subset to quantify the contribution of each FDGAN component and training choice. We consider: the full *FDGAN* (GAN active for timesteps $t < t_{\text{early}} = 400$ with ANPM and reconstruction losses), a variant with no GAN/ANPM (*No GAN/ANPM*, obtained by setting $t_{\text{early}} = 0$), a variant without reconstruction losses (*No $L_{recon}$*), a version with equal weighting of GAN and diffusion losses (*Equal Weighting*, $\lambda_{\text{GAN}}=1$), and a version with extended diffusion engagement (*Ext. Diffusion*, $t_{\text{early}}=4000$). We report three DINOv2-based metrics—Fréchet Distance (FD), Kernel Distance (KD), and FLS Overfit—and Inception-V3 FID. Lower is better for FD, KD, FLS Overfit (closer to $0$ indicates less overfitting), and FID. Removing

Table 2: Ablations on Cityscapes–Pedestrian (256×256). DINOv2 metrics (FD, KD, FLS Overfit) and Inception-V3 FID.

| Model Variant | FD↓ | KD↓ | FLS-O↓ | FID↓ |
|---|---|---|---|---|
| No GAN/ANPM ($t_{\text{early}}=0$, no $L_{\text{recon}}$) | 1276.70 | 5.78 | $-36.90$ | 91.89 |
| No $L_{\text{recon}}$ (GAN active) | 1366.19 | 6.22 | $-39.38$ | 105.30 |
| Equal Weighting ($\lambda_{\text{GAN}}=1$) | 1104.00 | 5.36 | $-33.31$ | 46.04 |
| Ext. Diffusion ($t_{\text{early}}=4000$) | 1142.91 | 5.18 | $-33.66$ | 55.61 |
| **FDGAN (full, $t_{\text{early}}=400$)** | **583.70** | **2.83** | **1.39** | **19.16** |

*both* ANPM and the GAN branch ($t_{\text{early}}=0$) degrades realism and generalization substantially (higher FD/KD/FID, more negative FLS Overfit), indicating that early-step adversarial feedback is pivotal when data are limited. Omitting $L_{\text{recon}}$ similarly harms fidelity and raises overfitting. Forcing equal GAN/diffusion weighting destabilizes the trade-off, and shifting adversarial engagement to much earlier timesteps (*Ext. Diffusion*) hurts quality. The full configuration (FDGAN) achieves the best overall balance across realism (FD/FID), distributional alignment (KD), and generalization (FLS Overfit closer to 0).

## 5 CONCLUSIONS

The FDGAN model introduced in this paper has demonstrated promising results in addressing the inherent challenges of object-centric image generation, particularly when confronted with limited or degraded training data. By integrating GAN adversarial training directly into intermediate steps of a diffusion model and applying targeted noise perturbations within bounding-box regions via the proposed ANPM, FDGAN appears to offer a viable solution to the common issue of overfitting under constrained dataset conditions. FDGAN demonstrated superior performance across most evaluation metrics compared to established generative models such as Projected GAN, LayoutDiffusion, Diffusion-GAN, and StyleGAN2-ADA. Specifically, it achieved lower Fréchet Distance (FD), Kernel Distance (KD), and optimal Coverage and Fidelity (CT and CT mod) metrics, reflecting enhanced perceptual quality and diversity. Furthermore, the lowest observed Feature Likelihood Score Overfit (FLS Overfit) underscores FDGAN's improved generalization capabilities, further supporting its practical applicability in real-world tasks.

Future work will focus on expanding the evaluation of FDGAN across more diverse and larger-scale datasets to further test its robustness, adaptability, and accuracy in object placement. Additionally, variations in discriminator architectures and extensions to multi-class scenarios will be explored to further enhance the model's versatility and broaden the scope of its practical applications.

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

## A  APPENDIX

## B  MODIFIED SAMPLING METHOD

In standard diffusion models (DMs), image generation begins from a fully noised image $x_T \sim \mathcal{N}(0, I)$ and proceeds through a sequence of reverse denoising steps to recover a clean image $x_0$. FDGAN modifies this process by injecting localized Gaussian noise within BB regions at sampling time. This simple but effective change focuses the model's attention on object-relevant areas during early denoising, improving placement and fidelity without requiring explicit conditioning.

Figure 3 illustrates the FDGAN sampling pipeline. It starts with an isotropic Gaussian noise image. Then, in block 2, an additional BB-targeted perturbation is applied. This localized noise is injected only into the BB-defined region, using a mask $M_{\mathrm{BB}}$ and a scalar multiplier $\gamma$, to amplify uncertainty in object regions. The perturbed image is then passed through the standard reverse diffusion loop guided by the U-Net generator.

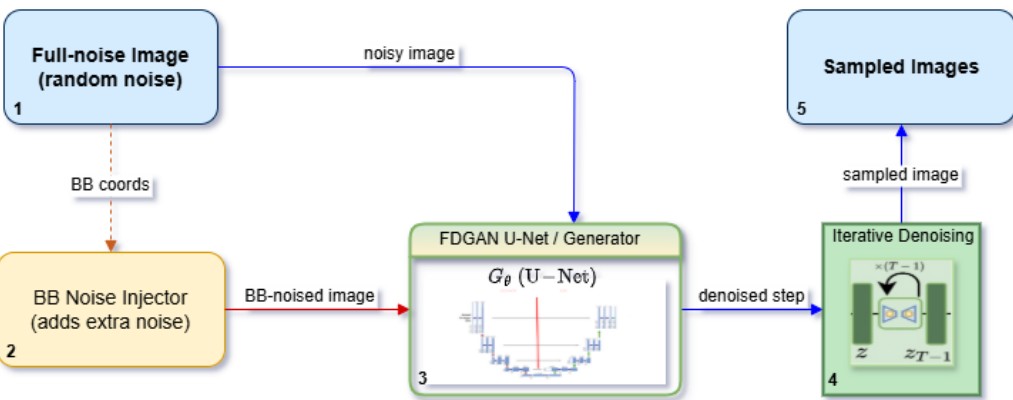

Figure 3: **FDGAN sampling pipeline.** Targeted noise is injected into object-specific BB regions before reverse diffusion begins.

**Sampling Pipeline.**  The steps are as follows:

- **(1) Initialization:** A full-noise image $x_T \sim \mathcal{N}(0, I)$ is sampled.
- **(2) BB Noise Injection:** Additional Gaussian noise is applied inside each BB region:

$$x_T' = x_T + \gamma \cdot M_{\mathrm{BB}} \cdot \epsilon, \tag{9}$$

  where $M_{\mathrm{BB}}$ is a binary mask (1 inside the BB, 0 elsewhere), $\epsilon \sim \mathcal{N}(0, I)$, and $\gamma$ is a hyperparameter controlling noise strength (set to 2 in our experiments). The background remains unaltered, ensuring that noise perturbations are confined to the object regions.
- **(3) Generator Input:** The BB-perturbed image $x_T'$ is passed to the U-Net generator $G_\theta$, along with the timestep $t = T$.
- **(4) Iterative Denoising:** Reverse diffusion begins, with the generator predicting noise $\hat{\epsilon}_\theta(x_t, t)$ at each step to estimate $x_{t-1}$. The model learns to progressively refine structure from high noise, particularly within BBs where uncertainty is greater.
- **(5) Output Sample:** After $T$ steps, a final image $x_0$ is produced.

**Motivation.**  This strategy implicitly encodes object localization through noise shaping rather than architectural conditioning. By increasing uncertainty inside BBs, the generator is encouraged to focus on those regions during early reverse steps—when coarse structure and semantic layout are established. Unlike layout-based models, this mechanism does not require feeding BB coordinates as inputs; the generator remains blind to BBs except through the noise pattern.

**Quantitative Use.** This modified sampling method is used consistently during evaluation. For all models trained with BB inputs, the same BB noise scheme is applied to test-time generation, ensuring fair comparison. Qualitative examples are shown in Fig. 4.

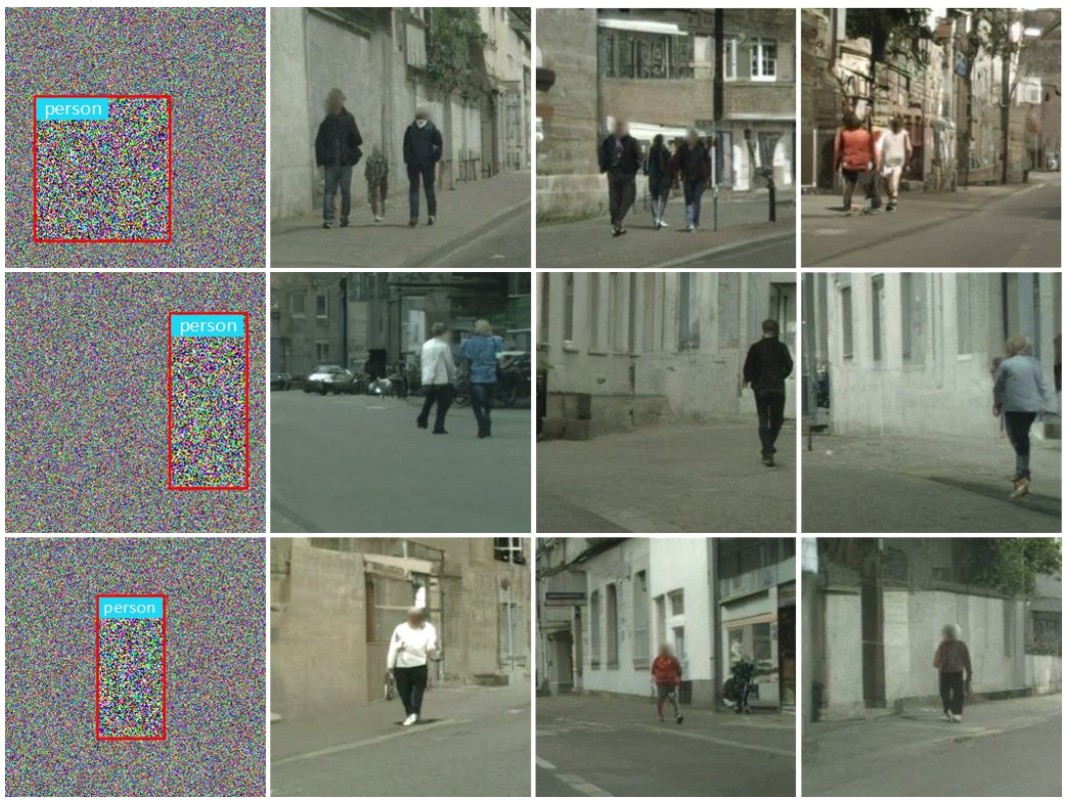

Figure 4: FDGAN samples generated from BB-perturbed noise. Objects emerge within designated regions, despite no explicit layout conditioning.

## C   DATASET

For FDGAN's evaluation, we curated three small, object-centric datasets (each $< 3k$ images) that span distinct scene complexities. From the high-resolution Cityscapes dataset (Cordts et al., 2016) (native frames $2048 \times 1024$), we extract a challenging *pedestrian* subset by retaining only the *person* class, obtaining BBs via a pretrained YOLOv5x detector (confidence threshold $> 0.8$, size $> 70 \times 32$ pixels). For each detection, we form a $256 \times 256$ crop—where the box is placed at a random offset within the crop—yielding dense, occlusion-heavy patches with privacy-blurred faces. The *Traffic-Signs* dataset comprises simpler, sparsely populated images obtained by cropping non-overlapping, sign-centered regions from $1024 \times 768$ street-view frames and resizing them to $256 \times 256$. Finally, an MS-COCO *potted plant* subset is cropped to $256 \times 256$, exposing diverse indoor/outdoor contexts. Evaluating FDGAN across these three settings probes robustness from dense occlusions (Cityscapes) and fine-detail small objects (Traffic-Signs) to semantically diverse object scenes with complex backgrounds (COCO), all under low-data constraints.

**Scope and pipeline overview.** The three datasets probe complementary challenges: dense, occlusion-rich scenes; simpler, well-isolated objects; and semantically diverse objects embedded in varied indoor/outdoor backgrounds. For all datasets we standardize to $256 \times 256$ crops, construct binary BB masks aligned with each crop, and apply synchronous transformations (when applicable) to images and masks. The Cityscapes preparation pipeline is illustrated in Fig. 5.

**Cityscapes Subset (Pedestrian Class):** From the original Cityscapes dataset, we extract all instances of the `person`, `rider`, `group` and `sitting person` classes, merging them into a unified `pedestrian`

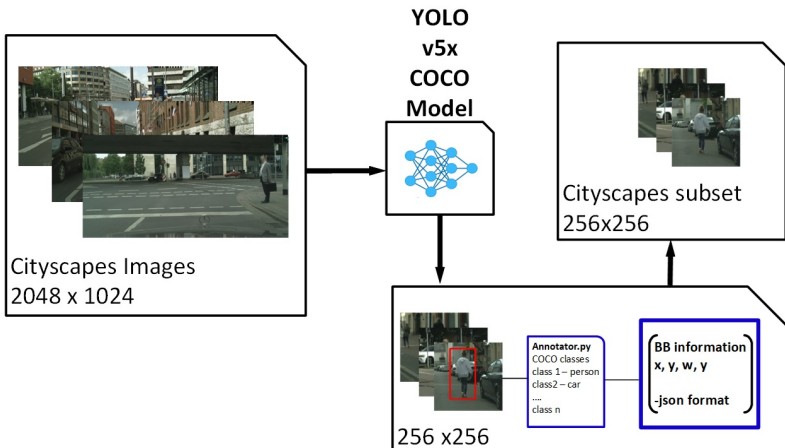

Figure 5: Dataset generation pipeline diagram.

category. BBs are generated using a YOLOv5x detector with a confidence threshold of 0.8, and minimum object size of 70×32 pixels. For each detection, a 256×256 patch is randomly cropped around the BB, ensuring variable object position and context within the image. This randomness forces the model to generalize beyond centered object placement. We collect 3,000 such samples for training.

This dataset presents several unique challenges:

- **Blurred Details:** All pedestrian faces are blurred for privacy, removing high-frequency visual features critical to photorealism.
- **Occlusions and Density:** Pedestrians frequently overlap or appear in groups, increasing ambiguity during training.
- **Feature Ambiguity:** Blurring and occlusion complicate the model's ability to learn consistent object structure.
- **Visual Artifacts:** The variability of blurred regions introduces nonuniform patterns that the model must learn to handle.
- **Ethical Compliance:** The dataset is privacy-compliant and adheres to ethical best practices when training on real-world scenes involving people.

**Traffic-Signs Dataset:** To create this dataset, we extract patches from 1024×768 street-view images containing clean, mostly centered Traffic-Signs. BBs are obtained via YOLOv5x, and we merge the `Traffic Sign` and `Damaged Traffic Sign` categories into a single class. A 256×256 patch is then cropped around each sign with randomized centering, ensuring positional variability. We again collect 3,000 images for training.

This dataset is substantially simpler than Cityscapes:

- **Single Object per Image:** Most samples contain one clearly isolated object.
- **No Occlusions:** The BBs are visually unobstructed and well-defined.
- **Visual Clarity:** Signs are typically high-contrast, centered, and free of noise or blur.

**COCO *potted plant* Subset (MS-COCO 2017, class id 58):** Starting from MS-COCO 2017 annotations, we select all images containing the `potted plant` class (category id 58). For each annotated instance we derive a $256 \times 256$ crop that *includes* the plant's bounding box at a random offset, so the object is not always centered and surrounding context varies. Concretely:

- **Instance selection:** We iterate over COCO `potted plant` instances and skip boxes that are too small to support a $256 \times 256$ crop without extreme upsampling (very small width/height or degenerate boxes).

- **Crop formation:** Given an instance box $(x, y, w, h)$, we sample a crop window of size $256 \times 256$ whose placement is randomly offset such that the annotated box lies fully inside the crop (with a small, uniformly sampled margin). When multiple instances are present, we generate one crop per instance (crops may overlap if plants are close).

- **Resolution and context:** Source images come from mixed native resolutions (e.g., many around $640 \times 480$) and both indoor/outdoor scenes, yielding diverse backgrounds and object appearances; all crops are standardized to $256 \times 256$.

- **Masks and alignment:** For each crop we rasterize a binary mask of the selected instance's BB into the crop's coordinates (1 inside the box, 0 outside). Images and masks undergo synchronous I/O and any dataset-level transforms to preserve alignment.

We cap the subset to fewer than 3k crops to match the low-data regime used throughout. This construction intentionally exposes background variability and object-placement diversity compared to the more uniform Traffic-Signs set, complementing the occlusion-rich Cityscapes subset.

## D  ANPM Details

Images with a timestep $t$ less than a predefined threshold $t_{\text{early}} = 400$ undergo an additional noising process performed by the ANPM. This module selectively applies extra Gaussian noise to specific spatial regions within the images, defined by BBs that encapsulate the target class objects. The noise is applied only to BB regions, leaving the background unchanged. Importantly, the magnitude of injected BB noise follows the diffusion model's native noise schedule for the current timestep $t$, and is further amplified by a constant factor $\gamma = 1.2$ to increase difficulty inside object regions. This guides the model to focus more on regenerating structure within those critical areas.

### D.1  Additional Noise Application Algorithm

---
**Algorithm 1:** Additional Noise Application to Partially Denoised Images

---
**Input:** Batch of images **micro**, conditional data **cond**, timesteps $t$
**Output:** Fake images with additional noise applied in BB regions
Initialize **bb_noise_masks** as tensor of zeros, same shape as **micro**;
**for** $j \leftarrow 1$ **to** *batch size* **do**
    Extract BB coordinates $(x, y, w, h)$ from **cond**$[j]$;
    actual_width $\leftarrow w - x$;
    actual_height $\leftarrow h - y$;
    noise_scale $\leftarrow$ diffusion.get_noise_scale_for_timestep($t[j]$);
    Generate Gaussian noise and apply to BB region:
    **bb_noise_masks**$[j, :, y : y + \text{actual\_height}, x : x + \text{actual\_width}]$ +=
      $\mathcal{N}(0, \text{noise\_scale} \times \gamma)$;
noised_images $\leftarrow$ diffusion.q_sample(**micro**, $t$, noise);
fake_images $\leftarrow$ noised_images + **bb_noise_masks**;
**return** *fake_images*

---

### D.2  Algorithm Details

The algorithm described above applies additional noise to partially denoised images before they are passed to the discriminator as fake samples. The detailed breakdown of each step is as follows:

**Initialization.** A tensor `bb_noise_mask` is initialized to zeros with the same shape as the input batch of images `micro`. This tensor will store the additional noise applied to the BB regions. **Processing Each Image in the Batch.** The algorithm iterates over each image $j$ in the batch. For each image, the BB coordinates $(x, y, w, h)$ are extracted from the conditional data `cond`. These coordinates define the region where the additional noise will be applied. The actual width and height of the BB are calculated as actual_width $\leftarrow w - x$ and actual_height $\leftarrow h - y$. **Calculating Noise Scale.** The noise scale for the current timestep $t[j]$ is calculated using the `get_noise_scale_for_timestep` method of the diffusion model. This value follows the model's native noise schedule and is further

multiplied by a fixed scalar factor $\gamma = 1.2$ to amplify perturbations inside the BBs. This factor is exposed as a hyperparameter. **Generating and Applying BB Noise.** Gaussian noise is generated for the BB region using the calculated noise scale. The generated noise is added to the corresponding region in the `bb_noise_masks` tensor. **Creating Noised Images.** The initial noise is added to the entire image using the `q_sample` method of the diffusion model, resulting in `noised_images`. The additional BB noise stored in `bb_noise_masks` is then added to these `noised_images`, creating the final `fake_images`. **Return Fake Images.** The final `fake_images`, now containing additional noise specifically applied within BB regions, are returned for further processing by the discriminator.

This targeted perturbation strategy ensures the discriminator receives informative inputs during early timesteps, enabling more precise feedback on object regions while maintaining coherent global structure.

# E    DETAILED TRAINING CONFIGURATION

Training for FDGAN was performed on two NVIDIA A100 GPUs (40 GB each), managed by SLURM with a 48 h time limit.

## E.1    CITYSCAPES AND TRAFFIC-SIGNS SETUP (BASE CONFIGURATION)

We use a two-stage procedure to integrate adversarial components stably. First, FDGAN is trained as a diffusion-only model (warm-up); subsequently, the *Additional Noise Perturbation Module (ANPM)* and the GAN branch are enabled. During adversarial training, only samples at *intermediate* denoising levels (measured by the reverse timestep) participate: for timesteps $t < t_{\text{early}}$, ANPM injects BB-localized noise and the discriminator evaluates the partially denoised outputs against real images. Unless otherwise noted, we set $t_{\text{early}}{=}400$.

**Generator and discriminator.**    The generator is a UNet with GroupNorm32 and SiLU activations, consistent with prior diffusion work (Nichol & Dhariwal, 2021). The discriminator is a PatchGAN N-layer network (Isola et al., 2017; Wang et al., 2018a).

Table 3: FDGAN configuration for Cityscapes/Traffic-Signs.

| Discriminator (PatchGAN / N-Layer) | |
|---|---|
| Architecture | 3 convolutional layers |
| Normalization | InstanceNorm |
| Activation | LeakyReLU (slope 0.2) |
| GAN loss | LSGAN |
| **Generator and Diffusion** | |
| Image resolution | $256{\times}256$ |
| Diffusion steps / schedule | 4000 / linear |
| UNet channels / resblocks | 128 / 2 |
| Attention / head channels | 32, 16 / 64 |
| ResBlock up/down; scale-shift | True; True |
| Learn sigma; dropout | True; 0.0 |
| **Training Hyperparameters** | |
| Warm-up | diffusion-only, then GAN+ANPM |
| GAN activation | samples with $t < t_{\text{early}}{=}400$ |
| Batch / micro-batch | 16 / 8 |
| Optimizers | AdamW (model), Adam (D/adapters) |
| Learning rates | $7{\times}10^{-5}$ (model), $1{\times}10^{-5}$ (D) |
| EMA rate | 0.9999 |
| Total iterations | $\sim 3.5{\times}10^{5}$ |
| ANPM noise multiplier | $\gamma = 1.2$ (BB-localized) |

*Notes.* (i) Only samples that are sufficiently denoised (here, $t < t_{\text{early}}$) are used for adversarial updates. (ii) For these datasets, the discriminator input is 3 ch RGB (no appended mask).

## E.2 COCO TRAINING VARIANT (REPRODUCIBILITY)

For the COCO *potted plant* experiments, we used a training variant tailored to small data with semantically diverse backgrounds. The FDGAN *core* is unchanged (GAN integration at intermediate steps with BB-localized perturbations); the following utilities differ from the base setup:

- **Late timestep band for adversarial updates.** Instead of a fixed $t_{\text{early}}$ gate, GAN/L1 updates are applied only when $t$ lies in a late band. We ramp $t_{\min}$ from 700 to 540 (band width 120) with $t_{\max}$ capped at 820; outside the band, diffusion-only updates are used. GAN warm-up is disabled.
- **ROI-focused noise and mask adapters.** For the GAN path, Region-of-Interest ROI-focused noise is injected with focus multiplier $k_{\text{focus}} \to 1.75$ and `noise_option=extra_gan_only` (an alternative `renorm_shared` is available; see code for details); discriminator inputs receive RGB+mask (4 ch) mapped to 3 ch via a $1\times1$ adapter.
- **Small ROI discriminator (optional R1).** Besides the main PatchGAN $D$, a small ROI $D$ (3 conv layers) is trained on mask-aligned ROIs; optional R1 on the small $D$ (e.g., `r1_gamma_small=5.0` every 16 steps) and a small integer translate diff-aug (`diffaug_-translate_px=4`) may be enabled.

**COCO-specific hyperparameters (concise).** *Unless noted, unspecified parameters follow Table 3.*

Table 4: Key deltas for the COCO variant.

| **Scheduler / banding** | |
| --- | --- |
| GAN band (late) | $t_{\min}$: $700 \to 540$, width 120, $t_{\max} \leq 820$ |
| GAN warm-up | disabled |
| **ROI utilities** | |
| Focus multiplier | $k_{\text{focus}} = 1.75$; `noise_option=extra_gan_only` |
| ROI crop / padding | `roi_size=128; roi_pad_ratio=0.03` |
| ROI area cap | `train_roi_frac_cap=0.15` |
| GAN samples per micro-batch | `max_gan_per_microbatch=4` |
| **Discriminator inputs** | |
| Main $D$ real source | `x_t`; small $D$ real source: `x0` |
| Small ROI $D$ | 3 conv layers; optional R1 (`r1_gamma_small=5.0`, `r1_every=16`) |
| DiffAug (small $D$) | `diffaug_translate_px=4` |
| **Base run settings (COCO)** | |
| Diffusion steps | 1000; mixed precision: `use_fp16=True` |
| Learning rate | $5\times10^{-5}$; batch / micro-batch: 16 / 8 |

**Sampling utilities.** For qualitative COCO samples we used a test-time ROI bump and optional PNPD (plug-and-play discriminator-guided) guidance within a narrow band (e.g., 140–205), reusing TorchScript-exported $D$/adapters (parameters such as `d_space`, `pnp_norm`, `pnp_gain`, etc.).

**Remark.** These changes affect *training utilities* only; the FDGAN core remains unchanged. Baselines are trained with the same BB masks for comparability.

**Robustness to box noise / absence.** In our target use cases (detector augmentation), BBs are available or can be pseudo-labeled. FDGAN tolerates moderate localization noise (empirically we found small jitter had limited impact on DINOv2 metrics); if boxes are unavailable, ANPM can be disabled to fall back to a generic hybrid diffuser (losing spatial focus), or BBs can be bootstrapped via off-the-shelf detectors. Exploring weak/learned saliency in place of hard boxes is a promising direction for future work.

### E.3 Computational overhead

We report wall-clock training time and parameter counts for the base setup ($256 \times 256$, 4000 steps) on $2 \times$ A100 40 GB; inference cost is unchanged because the discriminator is disabled at test time.

Table 5: Compute comparison ($2 \times$ A100 40 GB). Base: $256 \times 256$, 4000 steps; batch/micro-batch 16/8.

| Model | Params (M) | $\Delta$ vs. ADM | Train time | $\Delta$ vs. ADM |
|---|---|---|---|---|
| ADM (diffusion backbone) | $\sim$360 | – | $\sim$33 h | – |
| FDGAN (base, $t_{\text{early}} = 400$) | $\sim$370 | $+\sim$10 M ($<$3%) | $\sim$39 h | $+\sim$18% |

Inference overhead: none (discriminator unused at test time).
At $512 \times 512$, observed train-time increase is $\sim$23% (fits in 40 GB).

## F Dynamic GAN Loss Scaling

To effectively integrate GAN losses into the diffusion training process, we employ a dynamic scaling mechanism that modulates the GAN loss contribution based on the diffusion timestep $t$. This scaling algorithm progressively adjusts the weight of adversarial losses as denoising advances, amplifying their influence in later stages of image refinement. The detailed scaling procedure is described below:

---

**Algorithm 2:** Dynamic GAN Loss Scaling Algorithm

---

**Input:** Current timestep tensor $t$, maximum diffusion timestep $max\_t$, initial GAN loss scale $initial\_scale$, final GAN loss scale $final\_scale$
**Output:** Computed GAN loss scaling factor
Convert timestep tensor to floating-point: $t \leftarrow t.\text{float}()$;
Compute exponential decay rate:

$$\text{decay\_rate} \leftarrow \frac{\log\left(\frac{final\_scale}{initial\_scale}\right)}{max\_t}$$

Calculate scaled GAN loss factor:

$$\text{scale}(t) \leftarrow initial\_scale \times \exp(\text{decay\_rate} \times t)$$

**return** $scale(t)$

---

**Implementation and Integration Details.** At each training iteration, the current timestep $t$ determines the GAN loss scaling dynamically, ensuring adversarial guidance remains appropriately calibrated throughout the diffusion process. During early denoising stages (high noise levels, large $t$), the GAN loss scale factor remains small. This smaller weighting is critical at initial stages, as the model primarily concentrates on reconstructing broad image structures and managing substantial noise. During these initial stages, the diffusion-based reconstruction loss dominates, allowing the model to learn general image structure and global features without interference from potentially destabilizing adversarial gradients.

As training proceeds to intermediate and late denoising steps, the noise level decreases, and image content becomes partially clear. Here, the algorithm progressively increases the GAN loss scaling factor. The discriminator then provides targeted feedback on fine-grained details, textures, and object-level realism, precisely when the model is most receptive to these refinements.

The integration of this scaling factor directly modulates the generator's adversarial loss, complementing the diffusion loss and L1 reconstruction loss. This balanced, composite loss structure enables FDGAN to effectively harness GAN-driven realism exactly at the point in training when adversarial guidance is most beneficial.

*Implementation note.* We use the same schedule across experiments but apply it at different activation sets: (i) in the **base** setup (Cityscapes/Traffic-Signs), the schedule is applied only to samples that satisfy the early gate $t < t_{\text{early}}$; (ii) in the **COCO variant**, the schedule is combined with a late

timestep band, i.e., adversarial terms are applied only for $t \in [t_{\min}, t_{\max}]$, and within that band the per-sample diffusion loss is down-weighted (factor $0.7$) to balance objectives.

**Rationale for Dynamic Loss Scaling.** The primary rationale behind this dynamic loss scaling strategy is to smoothly transition training from an early-stage focus on diffusion-based noise removal and structural recovery toward a late-stage emphasis on GAN-driven detail refinement and image realism. Initially, when the image contains substantial noise, emphasizing GAN losses heavily could lead to instability and adversely affect model convergence. Conversely, applying limited or no GAN feedback near the end of denoising would sacrifice crucial fine-grained details and realism that adversarial training excels at capturing.

By dynamically scaling GAN losses, we carefully calibrate adversarial feedback according to the instantaneous denoising progress of the image. This ensures stable and effective training, where early reconstruction is primarily diffusion-driven, and later refinement incorporates powerful adversarial cues. As a result, the model achieves superior balance, yielding structurally coherent, highly detailed, and perceptually realistic object-centric images even under challenging limited-data scenarios.

*Sensitivity note.* Empirically, setting $t_{\text{early}}! \approx !0.1T$ (e.g., 400 of 4000) was most stable; shifts of $\pm 200$ steps mainly affected sharpness, whereas much smaller/larger values weakened gradients or increased artifacts.

## G  DINOv2 METRICS

### G.1  DISCUSSION OF EXTENDED METRICS

While the main paper focuses on the core metrics (FD, KD, FLS Overfit, and FID), the extended results reported in Table 6 provide a broader view of model behavior.

First, **Precision and Recall** highlight the fidelity–diversity trade-off. Across all three datasets, FDGAN consistently achieves higher recall than GAN-only baselines (e.g., ProjectedGAN, Pix2Pix), indicating better coverage of real data modes and reduced mode collapse. At the same time, FDGAN maintains competitive precision, demonstrating that improved diversity does not come at the expense of fidelity.

**Density and Coverage** provide complementary perspectives. FDGAN's density values are stable and close to those of the strongest baselines, suggesting it produces realistic samples that are not overly concentrated. Coverage values are consistently higher for FDGAN than diffusion-only models such as ADM or DiT, confirming that FDGAN balances both realism and distributional breadth.

The **Coverage Tests (CT and CT-mod)** further reveal that diffusion-only models tend to over-memorize under low-data conditions, while GAN-only models sometimes fail to generalize. FDGAN achieves CT values closer to zero, especially on Cityscapes and COCO, suggesting reduced memorization and stronger generalization.

Finally, $\text{FD}_\infty$, the bias-corrected variant of FD, supports the same ranking observed with FD. FDGAN maintains the lowest FD$\infty$ across datasets, reinforcing its advantage even after correcting for sample-size bias.

Taken together, these extended metrics provide consistent support for the view that FDGAN improves both diversity and generalization under limited-data conditions, while preserving fidelity. This consistency across independent measures strengthens the main-paper conclusion that integrating adversarial guidance at selective diffusion steps with BB-localized perturbations yields a more balanced generative model.

### G.2  FEATURE-SPACE COMPARISON: DINOv2 VS. INCEPTION-V3

We complement Inception-V3–based evaluation with DINOv2 encoder–based metrics, which provide a more comprehensive assessment of generative model performance. DINOv2 metrics have been shown to align better with human perception and capture a broader range of image characteristics (Stein et al., 2023; Jiralerspong et al., 2024).

Table 6: Extended metrics on Cityscapes, Traffic-Signs, and COCO *potted plant*. DINOv2 block (left): FD, $FD_\infty$, KD, FLS, CT, CT-mod, FLS Overfit. Inception-V3 block (right): FID, Precision/Recall, Density/Coverage. * denotes explicitly object-centric models or our modifications.

| | Class – Cityscapes Subset 256×256 – 3k | | | | | | | | | | | |
|---|---|---|---|---|---|---|---|---|---|---|---|---|
| **Encoder** | **DINOv2** | | | | | | | **Inception-V3** | | | | |
| Metric / Model | FD↓ | FD∞↓ | KD↓ | FLS↓ | CT | CT mod. | FLS overfit | FID↓ | Precision↑ | Recall↑ | Density↑ | Coverage↑ |
| **GANs** PSGAN* | 774.13 | 745.47 | 5.01 | 191.98 | 6.95 | 2.69 | −25.33 | 78.56 | 0.59 | 0.48 | 0.43 | 0.55 |
| Projected GAN | 828.32 | 791.10 | 3.98 | 118.67 | 9.86 | 12.51 | −11.93 | 15.81 | 0.60 | 0.80 | 0.52 | 0.79 |
| OC-ProjectedGAN* | 1076.03 | 1036.03 | 5.12 | 121.66 | 20.59 | 13.84 | −34.87 | 22.26 | 0.42 | 0.80 | 0.26 | 0.54 |
| Pix2Pix* | 958.23 | 929.03 | 4.27 | 197.40 | 16.63 | 2.76 | −27.64 | 88.78 | 0.41 | 0.40 | 0.22 | 0.36 |
| StyleGAN2-ADA | 1948.78 | 1912.15 | 8.77 | 164.99 | **4.85** | 9.62 | −48.73 | 71.36 | 0.28 | 0.28 | 0.12 | 0.17 |
| **DMs** ADM | 1275.56 | 1249.34 | 4.89 | 137.83 | 5.01 | 2.52 | −19.35 | 42.61 | 0.53 | 0.55 | 0.36 | 0.50 |
| DiT-XL/2-G | 2254.54 | 2219.05 | 11.64 | 180.70 | −10.35 | 17.61 | −49.13 | 92.65 | 0.12 | 0.15 | 0.03 | 0.04 |
| **FDGAN*** | **583.70** | **546.32** | **2.83** | **114.67** | −6.47 | **1.94** | **1.39** | 19.16 | **0.62** | **0.83** | 0.38 | 0.65 |
| Diffusion-GAN | 920.67 | 886.07 | 4.33 | 118.89 | 11.03 | 11.68 | −19.33 | **14.80** | 0.60 | 0.80 | **0.56** | **0.86** |
| Layout Diffusion* | 1313.52 | 1266.11 | 5.27 | 127.07 | 19.66 | 19.10 | −46.47 | 75.00 | 0.29 | 0.19 | 0.12 | 0.18 |

| | Class – Traffic-Signs 256×256 – 3k | | | | | | | | | | | |
|---|---|---|---|---|---|---|---|---|---|---|---|---|
| **Encoder** | **DINOv2** | | | | | | | **Inception-V3** | | | | |
| Metric / Model | FD↓ | FD∞↓ | KD↓ | FLS↓ | CT | CT mod. | FLS overfit | FID↓ | Precision↑ | Recall↑ | Density↑ | Coverage↑ |
| **GANs** Projected GAN | 586.91 | 560.09 | 2.24 | 247.12 | −7.12 | −33.07 | −42.6 | 18.18 | 0.60 | 0.62 | 0.56 | **0.68** |
| OC-ProjectedGAN* | 693.43 | 673.93 | 2.93 | 256.36 | **−0.56** | 33.96 | −43.66 | 33.66 | 0.32 | 0.72 | 0.17 | 0.28 |
| Pix2Pix* | 828.10 | 794.61 | 3.16 | 278.75 | −8.64 | 30.55 | −48.53 | 116.92 | 0.56 | 0.13 | 0.46 | 0.25 |
| StyleGAN2-ADA | 1487.25 | 1461.06 | 5.86 | 308.96 | −6.37 | −10.75 | 37.57 | 96.84 | 0.21 | 0.08 | 0.10 | 0.10 |
| **DMs** ADM | 551.36 | 527.99 | 2.18 | 126.09 | −20.63 | −14.52 | 33.20 | 34.76 | 0.59 | 0.64 | 0.50 | 0.56 |
| DiT-XL/2-G | 1349.91 | 1326.71 | 4.63 | 277.86 | −40.02 | 28.17 | −47.66 | 140.43 | 0.22 | 0.16 | 0.16 | 0.08 |
| **FDGAN*** | **416.19** | **391.95** | **1.54** | **177.94** | 7.04 | **5.52** | **−22.49** | 28.19 | **0.61** | **0.74** | 0.48 | 0.57 |
| Diffusion-GAN | 616.85 | 592.92 | 2.40 | 249.77 | −6.77 | 32.88 | −45.63 | **16.88** | 0.63 | 0.59 | **0.59** | 0.67 |
| Layout Diffusion* | 680.03 | 661.35 | 2.74 | 209.89 | 1.65 | 33.40 | −42.80 | 57.65 | 0.33 | 0.38 | 0.18 | 0.23 |

| | Class – COCO *potted plant* 256×256 – 2.3k | | | | | | | | | | | |
|---|---|---|---|---|---|---|---|---|---|---|---|---|
| **Encoder** | **DINOv2** | | | | | | | **Inception-V3** | | | | |
| Metric / Model | FD↓ | FD∞↓ | KD↓ | FLS↓ | CT | CT mod. | FLS overfit | FID↓ | Precision↑ | Recall↑ | Density↑ | Coverage↑ |
| **GANs** Projected GAN | 1094.78 | 1022.45 | 1.78 | 141.97 | 0.97 | 22.29 | -32.86 | 44.15 | 0.46 | 0.69 | 0.34 | 0.57 |
| OC-ProjectedGAN* | 1316.79 | 1253.33 | 2.52 | 144.53 | 2.63 | 21.99 | -33.06 | 61.75 | 0.40 | 0.64 | 0.25 | 0.40 |
| Pix2Pix* | 1623.55 | 1411.00 | 3.98 | 174.07 | -29.70 | -35.05 | 42.73 | 62.79 | 0.31 | 0.29 | 0.24 | 0.38 |
| StyleGAN2-ADA | 1450.62 | 1385.83 | 4.04 | 153.98 | -0.18 | 18.98 | -39.26 | 76.57 | 0.25 | 0.31 | 0.13 | 0.25 |
| PSGAN* | n/a | n/a | n/a | n/a | n/a | n/a | n/a | n/a | n/a | n/a | n/a | n/a |
| **DMs** ADM | 935.42 | 862.12 | 1.19 | 138.06 | -2.13 | 19.19 | -31.86 | 56.68 | 0.48 | 0.46 | 0.40 | 0.60 |
| DiT-XL/2-G | 926.54 | 843.84 | 1.53 | 126.64 | -0.30 | 20.02 | -30.93 | 56.10 | 0.45 | 0.69 | 0.25 | 0.33 |
| **FDGAN*** | **889.95** | **803.69** | **1.17** | **125.64** | -1.65 | 23.15 | **-28.93** | 43.83 | **0.58** | 0.69 | 0.39 | **0.62** |
| Diffusion-GAN | 1010.54 | 931.19 | 1.61 | 142.79 | 0.15 | 21.61 | -32.40 | **30.71** | 0.56 | **0.70** | **0.52** | 0.44 |
| Layout Diffusion* | 1047.62 | 963.50 | 1.62 | 140.81 | -1.31 | 23.38 | -33.93 | 62.95 | 0.44 | 0.49 | 0.31 | 0.52 |

Below, we summarize known limitations of the Inception-V3 model (Szegedy et al., 2015) commonly used in FID computation (Heusel et al., 2018), and then highlight the advantages of using DINOv2 (Caron et al., 2021) for evaluation.

**Training and Representation Issues:** The Inception-V3 network, traditionally used in FID computation, frequently fails to encode perceptually relevant features for datasets more complex than simple object-centric benchmarks such as CIFAR-10 or ImageNet (Kynkäänniemi et al., 2019). This limitation arises because Inception-V3 is trained explicitly for supervised classification on ImageNet, causing it to prioritize discriminative, class-specific features that may not generalize effectively across diverse image distributions (Stein et al., 2023). Consequently, metrics based on Inception-V3 often misalign with human evaluations of image quality, particularly for nuanced generative tasks, failing to accurately reflect subtle differences in realism or diversity (Naeem et al., 2020).

**Advantages of DINOv2:** In contrast, DINOv2 leverages self-supervised learning to extract semantically rich image representations without reliance on class labels (Caron et al., 2021). This training strategy enables DINOv2 to construct a more generalized and flexible representation space, capturing diverse image structures, textures, and semantic details more effectively than supervised counterparts such as Inception-V3 (Stein et al., 2023; Jiralerspong et al., 2024).

**Holistic Image Structure:** DINOv2 effectively encodes holistic image characteristics while simultaneously identifying and emphasizing key objects and their semantic context (Caron et al., 2021; Stein et al., 2023). This comprehensive representation ensures that important visual features are consistently captured, offering a richer and more robust evaluation framework for generative models.

**Better Alignment with Human Judgments:** Recent studies have demonstrated that the DINOv2 representation space aligns significantly better with human perceptions of image realism, diversity,

and quality compared to Inception-V3-based metrics (Stein et al., 2023; Jiralerspong et al., 2024). Consequently, metrics derived from DINOv2 feature embeddings more accurately reflect perceptual fidelity and diversity as evaluated by humans, thereby addressing a critical gap in generative model evaluation (Kynkäänniemi et al., 2019; Naeem et al., 2020).

**Self-Supervised Learning Benefits:** Unlike Inception-V3, which is constrained by supervised classification tasks, DINOv2 benefits from self-supervised learning paradigms that exploit vast quantities of unlabeled data (Caron et al., 2021). This enables it to generalize effectively across varying domains and image distributions, producing a representation space more adaptable and suitable for evaluating generative outputs.

**Increased Focus on Image Semantics:** DINOv2 emphasizes semantic content in images, facilitating more meaningful evaluations of synthesized images generated by diffusion and GAN models (Stein et al., 2023). Its semantic-centric approach ensures critical image aspects—such as contextual relevance, realism, and subtle perceptual details—are thoroughly captured and assessed. Given these advantages of DINOv2, we further illustrate its superiority through feature heatmap comparisons (Fig. 6). By generating heatmaps from both Inception-V3 and DINOv2, we visually demonstrate how each model interprets and represents image features. These heatmaps highlight differences in their focus and coverage, providing insights into their respective strengths and weaknesses in evaluating the quality and fidelity of generative outputs.

DINOV2                      Inception V3

Figure 6: **Heatmap comparison** between DINOv2 and Inception-V3. DINOv2 captures broader scene context, while Inception-V3 focuses narrowly on object-specific regions.

### G.3 METRIC DEFINITIONS

Below, we provide formal definitions and interpretations of the specific metrics we use in conjunction with the DINOv2 encoder:

**Kernel Distance (KD):** KD measures the Maximum Mean Discrepancy between real and generated image distributions in a high-dimensional feature space using a polynomial kernel, capturing differences in both distribution mean and variance (Bińkowski et al., 2021). *Lower is better.*

**Fréchet Distance (FD):** FD generalizes the original FID metric to DINOv2's feature space by comparing real and generated distributions modeled as multivariate Gaussians (Stein et al., 2023). *Lower is better.*

**FD∞ (Bias-Corrected FD):** FD∞ corrects the inherent sample-size bias present in the FD metric, providing an unbiased asymptotic measure of image distribution similarity (Chong & Forsyth, 2020; Stein et al., 2023). *Lower is better.*

**Coverage Test (CT):** CT evaluates whether generated images memorize or copy training samples by statistically comparing nearest-neighbor distances among training, generated, and test samples in feature space (Meehan et al., 2020). *Lower is better.*

**Modified Coverage Test (CT mod):** CT mod enhances the standard CT by reducing false positives and more reliably distinguishing genuine generalization from memorization (Stein et al., 2023). *Lower is better.*

**Feature Likelihood Score (FLS):** FLS assesses how likely generated samples are under the real-data distribution modeled in the feature space. It effectively balances realism, diversity, and novelty (Jiralerspong et al., 2024). *Lower is better.*

**FLS Overfit (Percentage of Overfit Gaussians, FLS-POG):** FLS Overfit quantifies the extent of memorization or overfitting, measuring how often generated samples are more similar to training samples than to unseen test data (Jiralerspong et al., 2024). *Closer to zero is better.*

**Precision:** Precision measures the proportion of generated samples close to real data manifold, thus quantifying image fidelity and realism (Kynkäänniemi et al., 2019). *Higher is better.*

**Recall:** Recall evaluates the coverage of the real distribution by generated samples, providing a direct measure of diversity and indicating mode collapse (Kynkäänniemi et al., 2019). *Higher is better.*

**Density:** Density refines precision by quantifying how densely generated samples populate the real data manifold, giving a nuanced view of realism and detail preservation (Naeem et al., 2020). *Higher is better.*

**Coverage:** Coverage complements recall by explicitly measuring the proportion of distinct real-data modes covered by generated samples, ensuring comprehensive representation and diversity (Naeem et al., 2020). *Higher is better.*

By employing these metrics, we provide a more comprehensive and nuanced evaluation of generative models. These metrics align better with human perception, capture a more complete structure of images, and leverage the advantages of self-supervised learning in DINOv2. This approach ensures that the evaluation of generative models is more accurate, fair, and reflective of their true capabilities in producing realistic and diverse images.

# H  ADDITIONAL QUALITATIVE RESULTS

This section presents example outputs from GAN-based, diffusion-based, and hybrid models (including FDGAN) across the three datasets considered in this work: Cityscapes–Pedestrian, Traffic-Signs, and COCO *potted plant*. For each dataset, we show two image grids: one for GAN-based baselines (FDGAN included for reference) and one for diffusion-based baselines (FDGAN included as well). These figures are intended as *illustrative* samples rather than a controlled qualitative study.

Samples were generated independently by each model using the evaluation settings described in the main paper; where applicable, bounding-box (BB) layouts or masks were provided to models that accept them. Because the compared methods differ in conditioning mechanisms (e.g., explicit BB layouts vs. unconditional generation), the displayed images are not matched on identical seeds or inputs. The grids thus serve to visualize typical artifacts and visual characteristics that accompany each approach under the low-data regime, complementing the quantitative metrics.

## H.1  CITYSCAPES RESULTS

**GAN-based methods.** Figure 7 shows representative samples from GAN-based models alongside FDGAN. Across methods, one can observe variations in human shape fidelity, textures, and background coherence characteristic of dense, occluded urban scenes. These examples are provided to illustrate the range of outputs produced by different models under identical dataset constraints.

**Diffusion-based methods.** Figure 8 presents examples from diffusion-based baselines and FDGAN. The images reflect common behaviors in this setting (e.g., blur vs. sharpness trade-offs, texture consistency, and background handling) and are included to complement the quantitative metrics reported in the main paper.

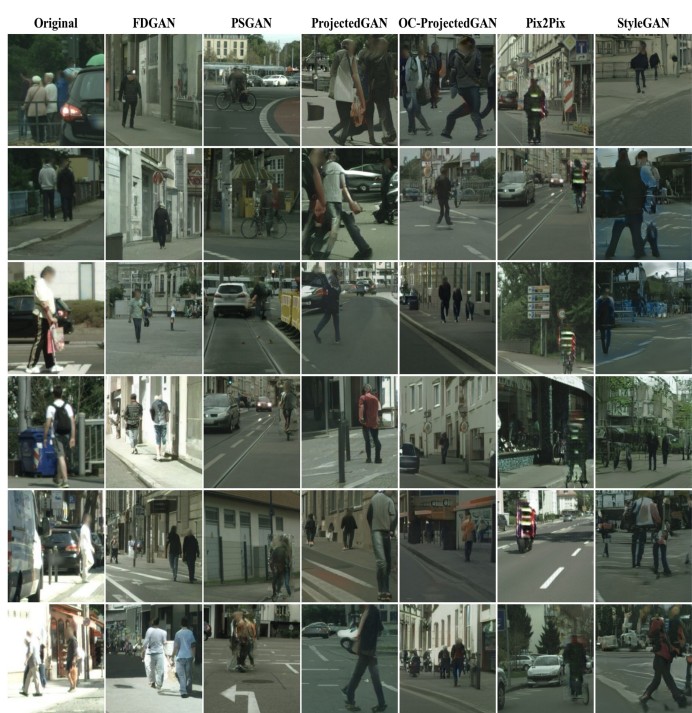

Figure 7: **Cityscapes–Pedestrian:** example outputs from GAN-based models. Columns show independent generations to visualize typical visual characteristics under low-data conditions.

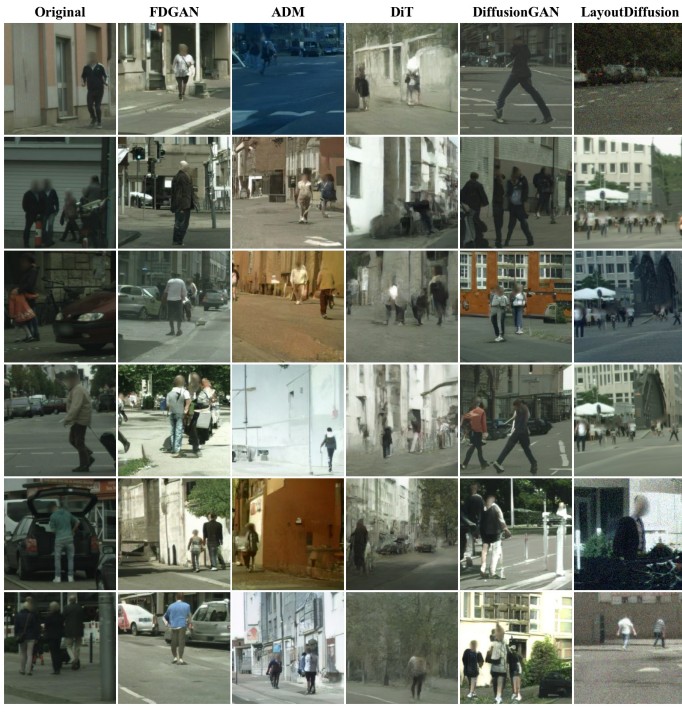

Figure 8: **Cityscapes–Pedestrian:** example outputs from diffusion-based models. Images illustrate typical results; no claim of qualitative superiority is implied.

## H.2 TRAFFIC-SIGNS RESULTS

**GAN-based methods.** Figure 9 shows examples from GAN-based models on the Traffic-Signs dataset. As a simpler, structured domain, typical variations include text legibility, edge sharpness, and the presence or absence of artifacts around sign boundaries.

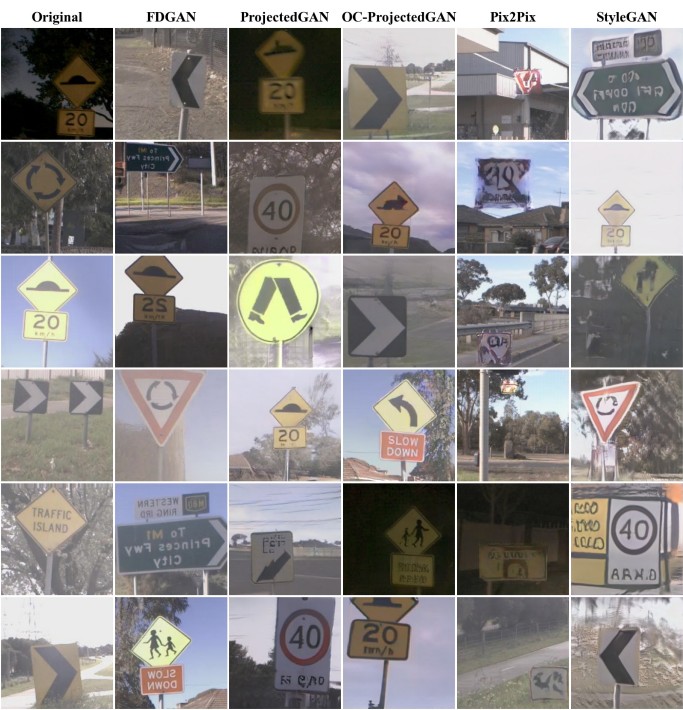

Figure 9: **Traffic-Signs:** example outputs from GAN-based models. Columns show independent generations illustrative of typical behaviors in this domain.

**Diffusion-based methods.** Figure 10 displays examples from diffusion-based baselines and FDGAN.

The samples illustrate characteristic outcomes under low data, including differences in small-text clarity, boundary smoothness, and background treatment.

## H.3 COCO *POTTED PLANT* RESULTS

**GAN-based methods.** Figure 11 shows examples on the COCO *potted plant* subset, which presents diverse backgrounds and object appearances (indoor/outdoor). The images illustrate model behaviors related to object–background compositing, leaf/branch detail, and overall scene coherence.

**Diffusion-based methods.** Figure 12 presents examples from diffusion-based baselines and FDGAN on the same subset. The samples visualize typical outcomes for fine structure (e.g., leaves), background handling, and object placement across varied scenes.

**Scope.** These qualitative figures are *illustrative* only and are not a substitute for a dedicated perceptual study. A systematic qualitative evaluation (e.g., human preference tests or protocolized blind ratings) is an interesting direction for future work. Here, the figures are intended to complement the quantitative metrics by providing visual context for typical outputs under the same low-data constraints. Because methods differ in conditioning mechanisms (e.g., explicit BB layouts vs. unconditional generation), seeds are not shared across models; instead, we fix a display protocol (panel order, number of samples, crop size) and keep it identical across datasets.

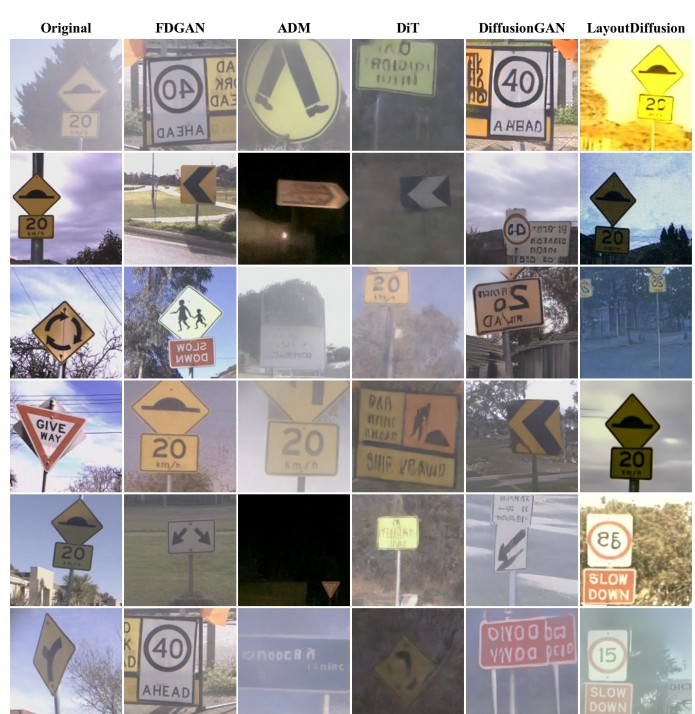

Figure 10: **Traffic-Signs:** example outputs from diffusion-based models. Images are representative and complement the quantitative comparisons.

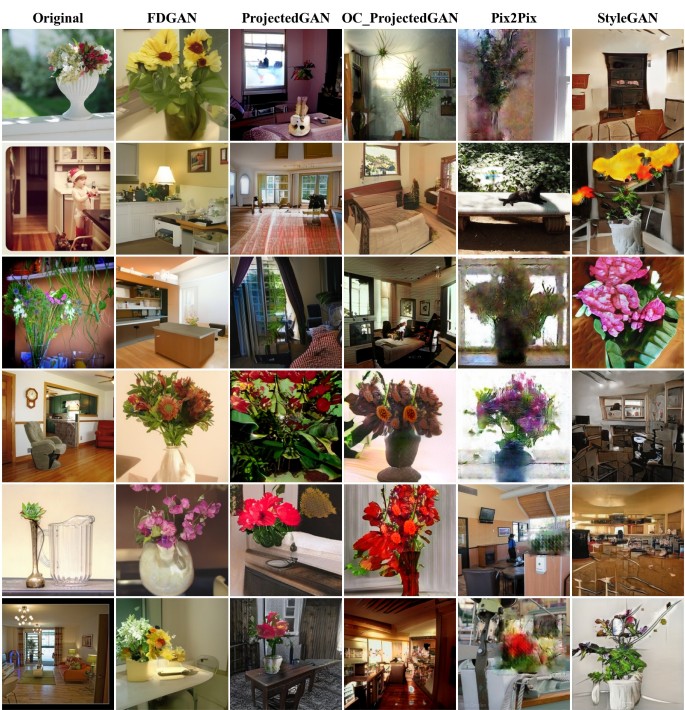

Figure 11: **COCO** *potted plant***:** example outputs from GAN-based models. Columns provide representative samples across varied contexts.

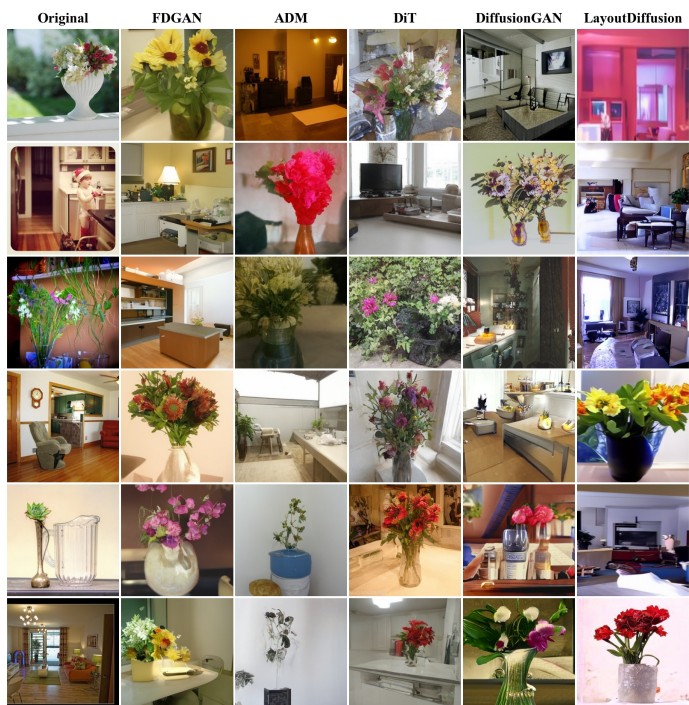

Figure 12: **COCO *potted plant*:** example outputs from diffusion-based models. These images are illustrative and not intended as a controlled qualitative study.

### H.4 ABLATION STUDY RESULTS

Figure 13 presents qualitative results from the FDGAN ablation study, clearly illustrating the impact of removing key model components. Specifically, we assess variants without the ANPM/GAN module,

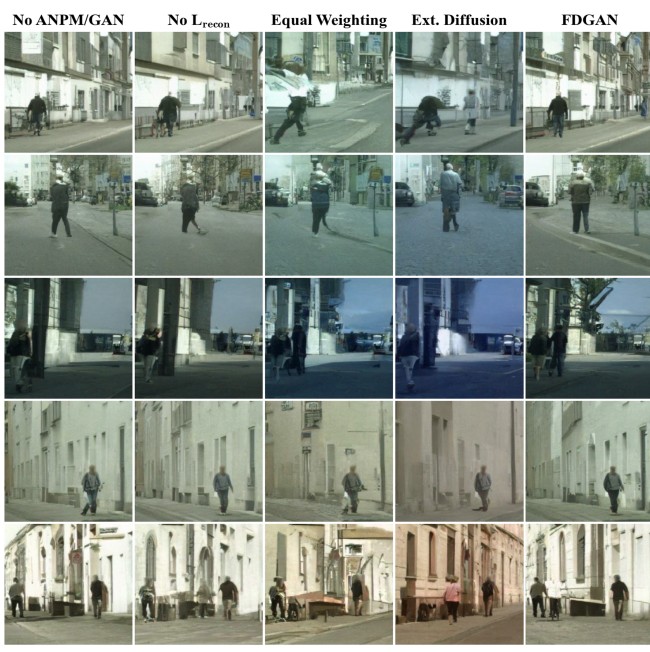

Figure 13: **FDGAN ablation study qualitative comparison.** Columns represent FDGAN variants illustrating the effect of various components.

reconstruction losses ($L_{\text{recon}}$), with equal weighting of GAN and diffusion losses, and with extended diffusion steps. Each ablation significantly degrades image quality, realism, and object-level detail preservation compared to the complete FDGAN model. The full FDGAN configuration (far right column) consistently yields the highest-quality images, highlighting the critical roles of targeted perturbations (ANPM), GAN feedback, and carefully balanced loss functions.

# I  OBJECT-CENTRIC ADAPTATION OF PROJECTEDGAN

We adapted ProjectedGAN (Sauer et al., 2021) into an object-centric conditional image generation framework (OC-ProjectedGAN) to provide an additional comparative baseline against FDGAN. This appendix details the exact code-level modifications and the intended implications. Evaluating how effectively OC-ProjectedGAN generates objects within prescribed bounding boxes (BBs) during inference is beyond the current scope, and thus not quantified here. Instead, we aimed solely to conditionally integrate bounding-box spatial constraints explicitly into ProjectedGAN.

**Generator Modifications.**  The original ProjectedGAN generator is unconditional, relying solely on latent noise $z$ for image synthesis. To introduce spatial guidance, the following modification was implemented in the generator:

Each input bounding-box mask, provided in YOLO-format annotations, is converted into a binary mask matching the target image dimensions (256×256). This binary mask is flattened and projected through a learned linear embedding layer to obtain a latent vector aligned with the original latent vector dimension:

$$z_{\text{cond}} = z + \text{Linear}(\text{flatten}(\text{BB\_mask})). \tag{10}$$

This *additive* conditioning method implicitly encodes spatial layout constraints directly into the latent representation before synthesis, enabling the generator to condition image generation explicitly on bounding-box annotations.

**Discriminator Modifications.**  To ensure the discriminator considers object placement, the following explicit spatial conditioning was implemented by augmenting the discriminator's input channels:

A binary bounding-box mask is concatenated directly with the RGB input images, resulting in a four-channel input tensor. Subsequently, this augmented input undergoes a single $1 \times 1$ convolutional layer to reduce channel dimensionality back to three channels compatible with the pre-trained discriminator backbone:

$$x_{\text{disc}} = \text{Conv}_{1\times1}(\text{concat}(x_{\text{img}}, \text{BB\_mask})). \tag{11}$$

This explicitly conditions the discriminator to evaluate both realism and spatial consistency, leveraging the provided bounding-box constraints.

**Dataset Preparation.**  The dataset was structured explicitly to pair each training image with its corresponding binary bounding-box mask derived directly from YOLO-format annotations. Pixels within bounding boxes were set to 1, and pixels outside were 0. All image augmentations (such as flips or crops) were synchronously applied to images and their associated masks to maintain precise spatial alignment.

**Inference Procedure.**  During inference, OC-ProjectedGAN uses the trained spatial conditioning mechanism as follows:

Given a noise vector $z$ and an externally specified bounding-box mask provided at test time, the generator synthesizes images conditioned explicitly on these spatial constraints. In practice, the bounding-box mask is loaded from a grayscale PNG file (or a directory of such masks). The generator's forward pass incorporates the mask embedding precisely as done during training:

```
img = G(z, label, truncation_psi, noise_mode, bb_mask=current_bb_mask)
```

**Scope and Limitations.** These targeted code-level modifications successfully adapt ProjectedGAN into a spatially-conditioned image generation model (OC-ProjectedGAN). However, this work does not include a rigorous quantitative analysis of the model's accuracy in strictly adhering to bounding-box placement during inference. Such an evaluation remains a compelling topic for future investigation.

In summary, OC-ProjectedGAN is explicitly designed and implemented as an object-centric model to complement comparisons with FDGAN, thus enriching our comparative analysis framework for object-centric generative models.

## J  FINE-TUNING PROTOCOL AND EARLY-STOPPING CRITERIA

**Pretrained initialization.** For small, object-centric datasets, initializing from author-released checkpoints is preferable to training from scratch: it leverages prior visual knowledge and consistently improves convergence and sample quality when the target data are scarce (Wang et al., 2018b; Grigoryev et al., 2022). We therefore fine-tuned each model on COCO *potted plant* starting from the corresponding public checkpoint.

**Checkpoint selection and stopping.** We evaluate generated samples every 20k optimization steps using a fixed evaluation protocol. For each snapshot we compute DINOv2 FD (primary), KD, FLS Overfit, and Inception-V3 FID, Precision, and Recall on a held-out set. We select the snapshot that *minimizes FD* subject to *maintaining Recall*, and stop training once these summary metrics plateau or begin to degrade. This choice mirrors common practice in generative modeling—selecting the best checkpoint by FID/precision–recall and terminating when further training yields diminishing returns or early signs of collapse (Heusel et al., 2018; Kynkäänniemi et al., 2019). When applicable, we also monitor diversity proxies (e.g., MS-SSIM (Wang et al., 2004)) to flag increasing redundancy.

**Qualitative guardrails.** Because losses alone are not reliable indicators of generative quality, we complement metrics with periodic qualitative checks on fixed seeds/prompts (unconditional and conditional settings, respectively). We halt before outputs become visually repetitive or backgrounds deteriorate, ensuring the chosen checkpoint captures the target concept while preserving diversity.

**Rationale in low-data regimes.** Stopping at the first snapshot where target-class fidelity is high and diversity remains intact reduces overfitting and mode collapse—failure modes that are amplified when fine-tuning on small datasets. This protocol is consistent with reports that pretrained generators retain broader coverage than scratch-trained models and benefit from shorter, carefully monitored fine-tuning schedules (Wang et al., 2018b; Grigoryev et al., 2022).

## LLM USAGE

We used an LLM (ChatGPT) only for minor copy-editing (e.g., wording, concision, and punctuation). All suggested edits were reviewed and approved by the authors, who take full responsibility for the final text.

