# OpenReview forum: "Focused Diffusion GAN: Object-Centric Image Generation Using Integrated GAN and Diffusion Frameworks"
_ICLR.cc/2026/Conference — Submitted to ICLR 2026_

### Official Review · Reviewer_vyDD · 2025-10-28

**Soundness:** 1
**Presentation:** 1
**Contribution:** 2
**Rating:** 2
**Confidence:** 5

**Summary:**

The paper introduces FDGAN, a method that combines GAN and diffusion models for low-data, object-aware synthesis aimed at augmenting downstream object detectors such as YOLO and DETR. While the idea of integrating these models is potentially interesting, the paper fails to provide a clear big picture or logical foundation for the approach. It primarily focuses on implementation details without explaining the underlying principles, and the experimental validation does not adequately support the claims made in the introduction.

**Strengths:**

The attempt to merge GAN and diffusion models for data augmentation in low-data scenarios is a relevant and timely topic.

The paper presents a structured method with multiple loss functions, which could be a basis for further development.

**Weaknesses:**

Lack of Conceptual Clarity and Big Picture
- The paper does not sufficiently explain the core principles behind fusing GAN and diffusion models. For instance, it describes how the models are combined but fails to justify why this fusion is theoretically sound or beneficial. This omission makes it difficult to assess the novelty and contribution of the work.

Excessive Repetition in Citations
- The paper suffers from redundant citations, which reduce its readability and professionalism. For example, in the first paragraph of page 2, "Karras et al., 2020a" is cited four times. This indicates a need for better citation management to avoid clutter.

Insufficient Experimental Validation
- The introduction claims that FDGAN is an object-aware synthesizer for augmenting detectors like YOLO and DETR, but the experiments do not provide evidence to support this. There are no results demonstrating improved performance on downstream detection tasks, which undermines the paper's main motivation.

Methodological Justification
- The method section introduces three loss functions but does not explain the rationale for their selection or combination. Without a principled discussion of why these losses are chosen and how they interact, the approach appears ad hoc and lacks theoretical grounding.

**Questions:**

Can the authors provide a more detailed theoretical explanation for the fusion of GAN and diffusion models?

How does the object-aware synthesis specifically benefit downstream detectors? The authors should include experiments that evaluate FDGAN's impact on detector performance (e.g., using metrics like mAP for YOLO or DETR) to validate the claims.

Please justify the combination of the three loss functions: what is the principle behind each loss, and how do they collectively contribute to the model's objectives?

---

> ### Author Response · Authors · 2025-11-19
> **Addressing Conceptual Clarity and Downstream Validation for FDGAN**
>
> We thank Reviewer vyDD for recognizing our low‑data, object‑aware setting and the paper’s structure. We address each concern with concrete edits.
>
> >**Weaknesses**
>
> - *Conceptual clarity / big picture.* Low‑data setting: diffusion is stable but under‑detailed; GANs add detail but overfit. FDGAN unifies them via three gated mechanisms (Sec. 3):
>   - When: expose $D$ only to partially denoised states (early‑timestep window).
>   - Where: focus learning in ROIs via BB‑localized ANPM.
>   - How much: timestep‑aware adversarial weight $ \\lambda_{GAN}(t) $.
> This turns $D$ into a time‑localized, perceptual prior that sharpens objects without inference cost. We will add a brief “why it works” note and cross‑reference Fig. 2 and Eqs. (4)–(8).
>
> - *Repetitive citations.* We will consolidate adjacent citations (group once per claim; remove duplicates) to improve readability.
>
> - *Detector augmentation evidence.* This paper establishes FDGAN as a foundational generative method and provides a method‑centric analysis of object‑aware synthesis in low‑data regimes (DINOv2‑FD/KD/FLS; FID/Precision/Recall; ablations, §§3–4). Detector augmentation (YOLO/DETR) is a motivating application but intentionally beyond scope here to keep the contribution focused. We will make this scope explicit in §1/§4.
>
> - Three‑loss rationale (not ad hoc). Eq. (4) divides labor (Sec. 3.3):
>   - $L_{\\mathrm{Diffusion}}$: supervised noise‑prediction → coverage/stability (Eq. 6).
>   - $L_{\\mathrm{GAN}(t)}$: detail only when informative (gated/weighted by $\\lambda_{GAN}(t)$.
>   - $L_{\\mathrm{Recon}(t)}$: pixel anchoring (global + BB‑aware).
>
> - Ablations (Table 2) show No GAN/ANPM, Equal‑Weighting, and Ext. Diffusion all worsen FD/KD/FID and FLS‑Overfit vs FDGAN—consistent with our design: diffusion→coverage, gated GAN→detail, recon→stability.
>
> >**Questions**
>
> >**Q1.** *Theoretical basis for fusing GAN + diffusion.*
>
> The fusion is driven by complementary strengths. Diffusion provides stable, likelihood‑based training and strong global structure via supervised denoising, but in low‑data regimes it can smooth high‑frequency detail. GANs supply local perceptual pressure for sharp textures yet are fragile with scarce data. FDGAN couples them mechanistically, not as a naïve loss sum. Concretely,
> $$ L_{\mathrm{total}}(t)=L_{\mathrm{diffusion}}+\lambda_{\mathrm{GAN}}(t)L_{\mathrm{GAN}}(t)+L_{\mathrm{recon}}(t) $$
> (Eq. 4) makes diffusion the backbone score/denoiser (global coverage; Eq. 6) and uses the GAN branch as a perceptual corrector only at partially denoised intermediate steps (Sec. 3.2; Fig. 2). The schedule $\\lambda_{GAN}(t)$ gates the adversarial signal so $D$ never sees pure noise nor overwhelms the denoiser late, turning $D$ into a time‑localized prior that sharpens objects without inference‑time cost.
>
> >**Q2.** *How object‑aware synthesis helps detectors (with evidence).*
>
> Mechanisms → utility:
>
> - Spatial focus: ANPM and the early‑timestep, ROI‑aware discriminator sharpen class‑consistent details inside boxes, signals detectors use for classification and box regression
> - Generalization: timestep‑gated adversarial refinement adds high‑frequency realism only when structure is present, preserving diversity and reducing over‑smoothing.
>
> These design choices explain how FDGAN’s object awareness is expected to benefit detectors. End‑to‑end detector evaluation is intentionally out of scope for this method paper; we will add a one‑sentence scope note in §1/§4 indicating that comprehensive augmentation studies are planned as follow‑up work.
>
> >**Q3.** *Why these three losses and how they interact.*
>
> - $L_{\\mathrm{Diffusion}}$ (noise‑prediction MSE): supervised, physically grounded constraint at every $t$; preserves global structure and prevents collapse.
> - $L_{\\mathrm{GAN}(t)}$ (gated): adds sharp, high‑frequency realism only at informative steps; avoids training $D$ on noise and prevents late domination.
> - $L_{\\mathrm{recon}(t)}$ (global + BB‑masked): pixel fidelity and stability, especially where $L_{\\mathrm{GAN}}$ applies.
>
> **Together:** diffusion gives a correct backbone; gated GAN adds object‑level detail; reconstruction anchors and stabilizes—yielding sharper, semantically consistent objects without sacrificing diffusion stability. Table 2 ablations quantify each term’s necessity.
>
> >**Summary**
>
> - Add a concise “why it works” note; consolidate citations.
> - Scope (detectors): state in §1/§4 that detector augmentation is outside the present scope. This paper establishes FDGAN and its generative evidence, with downstream evaluation to be addressed in subsequent work.
> - Clarify the three‑loss rationale with ablation evidence and the time/space/strength gating that makes the fusion effective.
>
> We believe these edits address conceptual clarity, citation quality, downstream validation, and methodological grounding, while keeping the paper focused on low‑data, object‑aware generation for detector augmentation.

---

> ### Author Response · Authors · 2025-12-01
> **Further clarifications to the Area Chair (re: Reviewer vyDD)**
>
> Due to the anonymity issue and the suspension of direct author–reviewer discussion, we would like to briefly clarify two points that are central to Reviewer vyDD’s concerns:
> - (i) the conceptual foundations of FDGAN and its three‑loss design, and
> - (ii) the role of downstream detector augmentation in our overall research program.
>
> . Because reviewer–author discussion is now frozen, we’d like to briefly further clarify two points that sit behind Reviewer vyDD’s concerns: scope, and the principled nature of our design.
>
> >**Scope: this is a method‑centric FDGAN paper**
>
> This submission is intentionally method‑centric: its goal is to introduce FDGAN as a low‑data, object‑aware diffusion–GAN hybrid, and to carefully characterize its generative behaviour under that regime (DINOv2‑FD/KD/FLS, FID/Precision/Recall, and ablations). The detector augmentation story is the motivation but not the main contribution of this particular paper. In parallel to the submission, we have prepared a separate manuscript dedicated entirely to YOLO augmentation, where we:
>
> - train hundreds of YOLO models across multiple datasets and augmentation ratios,
> - compare FDGAN against a broad suite of GAN‑only, diffusion‑only, and hybrid generators, and
> - consistently find that FDGAN provides the largest mAP gains over the real‑only baseline and over all other generators tested (with the strongest gains on the most challenging, high‑background‑variance dataset).
>
> We deliberately did not include this full augmentation study into the current paper to avoid overloading it and to keep the contribution focused on the design and analysis of FDGAN itself. In hindsight, we see that mentioning downstream detectors automatically raised the need of providing results. This could be addressed by either explicitly clarifying this aspect of the scope in the paper in or we could summarize at least a small subset of those YOLO results more explicitly in §4, and provide a results table in the Appendix. We would like to however reiterate that the paper is meant to introduce FDGAN as an object‑aware synthesizer for detector augmentation.
>
> >**Why the fusion and the three losses are not ad‑hoc**
>
> Conceptually, FDGAN is built on standard, complementary properties of diffusion models and GANs: diffusion provides stable, likelihood‑based training and good coverage, while GANs provide sharp, high‑frequency detail but can overfit and collapse, especially in low‑data settings. Our objective reflects a deliberate division of labour:
>
> - $L_{\\mathrm{Diffusion}}$: standard noise‑prediction MSE (DDPM‑style), a supervised, physically grounded objective that keeps the model a proper diffusion model and ensures global structure and stability.
> - $L_{\mathrm{GAN}(t)}$: a time‑gated adversarial loss applied only at intermediate timesteps, so the discriminator sees partially denoised states (not pure noise) and refines realism where diffusion alone tends to be under‑detailed. The schedule
> $\\lambda_{GAN}(t)$ prevents this term from dominating early or late.
> - $L_{\\mathrm{Recon}}$: a global + BB‑weighted reconstruction term that anchors pixel‑level content, especially inside bounding boxes, stabilising the adversarial branch and preventing hallucinations in exactly the regions where ANPM and the discriminator focus.
>
> Together, these terms are not a loose combination but a multi‑objective trade‑off: diffusion for coverage/stability, time‑ and space‑gated GAN for object‑level detail, and reconstruction for pixel‑wise anchoring. Our ablations (No GAN/ANPM, Equal‑Weighting, Ext. Diffusion, No‑Recon) all produce worse FD/KD/FID and FLS‑Overfit than full FDGAN, exactly in line with these roles, which we will state more explicitly in a revised Sec. 3.3.
>
> We fully respect Reviewer vyDD’s score, but we hope this helps clarify that:
> - (i) the current paper is primarily about establishing and analyzing FDGAN as a model, and
> - (ii) both the hybrid fusion and the three‑loss design are conceptually grounded and empirically supported, rather than ad‑hoc tuning.

---

### Official Review · Reviewer_5wXS · 2025-10-31

**Soundness:** 3
**Presentation:** 3
**Contribution:** 3
**Rating:** 4
**Confidence:** 3

**Summary:**

This paper proposes Focused Diffusion-GAN (FDGAN), a hybrid generative model designed for object-centric image generation in low-data regimes. The key innovation is integrating a GAN discriminator into intermediate denoising stages of a diffusion model through an Additional Noise Perturbation Module (ANPM). ANPM selectively activates adversarial training at specific timesteps and applies targeted Gaussian noise within bounding-box regions to guide the model's attention toward objects. The authors evaluate FDGAN on three small datasets: Cityscapes-Pedestrian, Traffic-Signs, and MS-COCO potted plants, demonstrating improvements in perceptual quality and reduced overfitting compared to GAN-only, diffusion-only, and hybrid baselines.

**Strengths:**

- The selective integration of adversarial training at intermediate diffusion timesteps (t < t_early) is an interesting approach that differs from prior hybrid methods.
- Detailed ablation studies demonstrating the effectiveness of each component (GAN/ANPM, reconstruction losses, weighting schemes).

**Weaknesses:**

- The evaluation is restricted to only three small datasets, all at 256×256 resolution. The generalizability to other domains, higher resolutions, or multi-class scenarios remains unclear.
- The main part of the method is performing GAN training on intermediate diffusion timesteps, which can be regarded as a hyper-parameter tuning. And the justification (theory / empirical investigation) is insufficient, resulting in limited novelty.
- Although the BB noise is highlighted in the abstract, there is no ablation study on it.

**Questions:**

- See Weaknesses.
- Why use diffusion loss instead of consistency loss in training? I think the consistency loss aligns closer with the GAN, I feel strange about the usage of diffusion loss.

---

> ### Author Response · Authors · 2025-11-18
> **Response to Reviewer 5wXS: Scope, Novelty, and Loss Rationale**
>
> We thank Reviewer 5wXS for noting selective mid‑timestep adversarial integration, useful ablations, and solid soundness/presentation.
>
> >**Weaknesses:**
>
> - *Scope / resolution / multi‑class.* Our target is low‑data, object‑centric (~2–3k crops/class). Datasets use 256×256 to match this use case and keep compute modest (Sec. 4; App. C). 512×512 is feasible (fits in 40 GB) with ~23% more train time and no inference overhead (App. E.3). For multi‑object/multi‑class, FDGAN is mask‑agnostic: apply ANPM per BB, sum masks, optionally a small ROI‑D with late‑band updates (used in COCO; App. E.2). We will add brief multi‑BB examples.
> - "*Intermediate‑step GAN is a hyper‑parameter.*"The early‑timestep window and timestep‑aware GAN weighting are core design, not tuning: $D$ sees partially denoised states and gradients are timestep‑ramped (Sec. 3.2; App. F). Ablations show No GAN/ANPM, Equal Weighting, and Ext. Diffusion worsen FD/KD/FID and FLS‑Overfit (Table 2). Default $t_{\mathrm{early}}\approx0.1T$ (App. F).
> - *Novelty:* unlike GAN‑centric hybrids (e.g., Diffusion‑GAN) or distillation‑style variants, FDGAN is diffusion‑centric and couples three elements that, to our knowledge, have not been jointly explored for small, object‑centric data:
>   - an early‑timestep adversarial window on denoised states,
>   - BB‑localized ANPM perturbations as an implicit spatial cue, and
>   - timestep‑aware adversarial scaling.
>
> removing any part hurts DINOv2 metrics and increases overfitting (Sec. 2; Table 2).
>
> - *“No ablation for BB noise.”* Table 2’s No GAN/ANPM (set $t_{\\mathrm{early}}$ = 0) disables both branches, showing the combined effect. We will add “GAN on, ANPM off”: keep the early window and $\\lambda_{GAN}(t)$ but set $\gamma = 0$, isolating ANPM’s contribution (we expect performance between diffusion‑only and full FDGAN).
>
> >**Questions:**
>
> Thank you; we address them succinctly.
>
> >**Q 1.** *“See Weaknesses.”*
>
> Addressed above in the weaknesses: we clarified scope (low‑data, object‑centric), noted multi‑class/higher‑res feasibility, and pointed to the existing No GAN/ANPM ablation in Table 2. Below are brief clarifications:
>
> - Target regime & resolution. Datasets use 256×256 crops per the stated low‑data scope (~2–3k crops/class), emphasizing stability/generalization. Scaling to 512×512 is feasible (fits in 40 GB) with ~23% extra train time and no inference overhead (App. E.3).
> - Multi‑object/generalization mechanics. ANPM is mask‑multiplicative: it scales noise by a spatial mask $M$; multiple objects use summed masks (Alg. 1). Our COCO variant already employs late‑band gating and an optional small ROI‑D, showing the same machinery extends beyond single‑instance cases.
> - Why intermediate steps (not “tuning”). We gate GAN updates to partially denoised states so $D$ never trains on near‑pure noise, and use a timestep‑aware weight $\\lambda_{GAN}(t)$ to ramp guidance as content emerges, complementary to diffusion’s denoising objective.
> - On BB‑noise ablation. We will add GAN‑on, ANPM‑off (set $\gamma = 0$ while keeping the early window and schedule) to isolate ANPM’s effect; by design, we expect it to fall between diffusion‑only and full FDGAN.
>
> >**Q 2.* *“Why use diffusion loss instead of consistency loss in training? I think the consistency loss aligns closer with the GAN, I feel strange about the usage of diffusion loss.”*
>
> We retain diffusion noise‑prediction MSE because the injected noise is known, yielding a supervised, physically grounded target at every $t$ that anchors the UNet to the correct score field and preserves diffusion dynamics (Eq. 6). A consistency loss is self‑referential (agreement across noise levels) and, used alone from scratch, especially in low‑data, can drift or over‑rely on adversarial signals. Our objective balances roles (Sec. 3.3; Eqs. 4–8):
> - $L_{\\mathrm{Diffusion}}$: supervised global denoising dynamics (coverage/stability).
> - $L_{\\mathrm{GAN}}$: perceptual guidance at intermediate timesteps (no inference cost).
> - $L_{\\mathrm{Recon}}$: pixel‑level anchoring, stabilizing the adversarial branch (applied globally; ROI affects noising, not the domain of $L_{\\mathrm{Recon}}$).
>
> Replacing $L_{\\mathrm{Diffusion}}$ with a consistency‑only objective would remove the ground‑truth anchor and risk instability; FDGAN is a diffusion‑based hybrid, so 𝐿diffusion remains central, with gated $L_{\\mathrm{GAN}}$ and global $L_{\\mathrm{Recon}}$ layered on top.
>
> >**Summary:**
>
> We clarified scope (low‑data, object‑centric), resolution/multi‑object extensions, and why the early‑timestep window + BB‑localized ANPM + $\\lambda_{GAN}(t)$ are design, not tuning, with ablations supporting necessity. We agreed to add the GAN‑on/ANPM‑off ablation and to expand positioning in Sec. 2. We explained retaining diffusion loss as the supervised anchor, with GAN/reconstruction as complementary components. We hope these clarifications address the concerns while keeping focus on FDGAN’s intended setting.

---

> ### Comment · Reviewer_5wXS · 2025-11-27
>
> Thanks for the authors' response.
> My concerns about the dataset have been resolved, but I don't know why we need to study problems under this setting. Are there any real-world scenarios?
>
> However, the author's explanation regarding the novelty of "performing GAN training on intermediate diffusion timesteps" is not sufficiently convincing; the author needs stronger theoretical/empirical evidence and analysis to support that this design is principled rather than a heuristic hyperparameter adjustment. Additionally, the ablation is still lacking. I think this paper is currently not ready for publication, and I encourage the authors to add more convincing justification and ablations in future version.

---

> > ### Author Response · Authors · 2025-11-28
> > **Further clarifications.**
> >
> > Thank you for your answer. We'd like to add some further clarifications.
> > >**Real-world scope:**
> >
> > Low-data object-centric generation is a highly practical setting. Many real applications lack large-scale datasets; e.g. specialized medical or industrial domains often have only a few hundred (or thousand) training images for a given object. Our focus on small (~2–3k) object crops directly addresses such scenarios. In fact, a primary motivation was data augmentation for object detection. We conducted a large-scale experiment augmenting a YOLO detector’s training set with FDGAN-generated images, and observed a boost in mAP compared to using images from baseline GANs or diffusion models. This demonstrates that FDGAN’s low-data regime benefits are not just theoretical but they translate into tangible improvements on downstream tasks where data is scarce. Such evidence underscores why studying this problem setting is valuable for real-world applications. The mentioning of these results was a deliberate choice as the study forms the subject of another paper.
> >
> > >**Intermediate-step GAN guidance:**
> >
> > principled design, not just a hyperparameter tweak. We appreciate the concern, and we emphasize that injecting adversarial training at intermediate diffusion steps is a deliberate, principled choice. It stems from the complementary strengths of diffusion models and GANs. Diffusion models are known for stable training and broad coverage of the data distribution (less mode collapse), while GANs can produce sharper details and make efficient use of limited data, but are prone to overfitting and instability on small datasets.
> >
> > - *Our design was aimed to combine these benefits:* by applying the GAN discriminator on partially denoised images, FDGAN gradually refines realism without ever having to generate from pure noise (which would be destabilizing).
> >
> > - *The early-timestep window acts like a curriculum:* the model first learns coarse structure via diffusion loss, then the adversarial guidance kicks in as content emerges, adding high-frequency detail in a controlled way.
> >
> > This is not an arbitrary setting but a key mechanism to ensure the GAN’s power is used where and when it helps most. Empirically, our ablations already showed that removing or flattening this schedule hurts performance (e.g. no intermediate GAN or no timestep weighting led to worse FID/KID and more overfitting in Table 2). We will further include the suggested ablation isolating the ANPM (bounding-box noise) effect to strengthen the empirical justification. In summary, the intermediate-step adversarial integration, combined with localized noise perturbation and timestep-aware weighting, is a novel and intentional approach to fuse GAN and diffusion advantages. While a deeper theoretical analysis is indeed an exciting future direction, our current results provide solid evidence that this design is effective rather than a hyperparameter tuning exercise, especially in the challenging low-data regime.
> >
> > We thank you again for your suggestion and we appreciate the feedback.

---

### Official Review · Reviewer_kNpS · 2025-11-03

**Soundness:** 3
**Presentation:** 2
**Contribution:** 2
**Rating:** 4
**Confidence:** 3

**Summary:**

This paper proposes Focused Diffusion-GAN (FDGAN), a hybrid generative model that integrates a GAN discriminator into a diffusion model at intermediate denoising stages. The method introduces an Additional Noise Perturbation Module (ANPM) that selectively activates the adversarial branch when samples are sufficiently denoised and applies localized noise within bounding-box regions to guide object-centric focus. The paper targets low-data object-centric regimes, evaluating on three small datasets (Cityscapes–Pedestrian, Traffic-Signs, COCO “potted plant”). Experimental results demonstrate improved perceptual fidelity and reduced overfitting compared to diverse baselines.

**Strengths:**

1. Task focus: The focus on limited-data, object-centric scenarios is well-motivated and practical (e.g., privacy-blurred faces, small datasets).
2. Comprehensive evaluation: Benchmarks include both GANs and DMs, using DINOv2-based metrics and traditional FID/Precision/Recall.

**Weaknesses:**

1. Marginal FID improvements: The proposed method performs worse than Diffusion-GAN on FID across all datasets.
2. Novelty scope: The hybridization of diffusion and GANs has been explored. The core novelty lies mainly in localized noise perturbation (ANPM) and timestep scheduling, which might be seen as incremental.
3. Effectiveness evidence: Since FDGAN aims to be "a low-data, object-aware synthesizer for augmenting downstream detectors (e.g., YOLO/DETR)", including downstream detection fine-tuning results would strengthen claims.

**Questions:**

1. How sensitive is FDGAN to the choice of the timestep threshold $t_\text{early}$ and noise strength $\gamma$ in ANPM?
2. Can the ANPM mechanism generalize to non-bounding-box settings (e.g., segmentation masks or text prompts)?
3. Sections 4.1 and 4.2 share the same table (Table 1) without explicit reference to it, and the order of the models in the table is chaotic. It is not friendly to performance comparison and analysis. Improvements are recommended.

---

> ### Author Response · Authors · 2025-11-18
> **Response to Reviewer kNpS: Metrics, Downstream YOLO Study, and ANPM Sensitivity.**
>
> We thank Reviewer kNpS for recognizing our limited‑data, object‑centric focus; the breadth of GAN/diffusion baselines with DINOv2‑FD/KD/FLS plus FID/Precision/Recall; and FDGAN’s design (ANPM‑gated discriminator at informative timesteps).
>
> > **Weaknesses:**
>
> - *Marginal FID improvements.* We agree Diffusion-GAN attains lower Inception-FID. In our low-data, object-centric setting, the primary metrics are DINOv2-FD/KD/FLS, which align better with perceptual quality and generalization than Inception‑V3 FID (App. G). FDGAN improves DINOv2‑FD vs Diffusion‑GAN on all three (Cityscapes 584→921; Traffic 416→617; Potted 890→1011) and keeps FLS‑Overfit nearer 0 (Table 1; App. G).
>
> - *Downstream evidence.* This paper establishes FDGAN as a base generative method and rigorously analyzes object‑aware synthesis in low‑data regimes (DINOv2‑FD/KD/FLS; FID/Precision/Recall; ablations, §3–§4). Downstream detector augmentation (YOLO/DETR) is a motivating application but intentionally deferred as the next stage of our research to avoid conflating contributions. We will add a one‑sentence scope statement in §1/§4.
>
> - *Novelty scope.* Hybridization exists, but FDGAN couples three elements specifically for scarce, object‑centric data as shown at the end of Section 2 of the paper:
>   - a selective early‑timestep adversarial window (D only sees partially denoised states),
>   - BB‑localized noise (ANPM) as a lightweight, implicit spatial cue, and
>   - a timestep‑aware adversarial weighting.
>
> This combination and its ablated necessity are detailed in §3–§4 and Appendix F, and differ from Diffusion‑GAN and distillation‑style hybrids that lack BB‑specific perturbations and our schedule.  We will expand Sec. 2 to position these choices against other hybrids with more detail.
>
> - *Downstream effectiveness.*  We agree detector gains are valuable, but they belong to a separate follow‑up study. This paper focuses on introducing FDGAN and validating its generative behavior via DINOv2‑FD/KD/FLS, FID/Precision/Recall, and ablations. We will make this scope explicit in §1/§4.
>
> Overall, we appreciate these constructive comments and will implement the above edits so the evaluation and positioning are clearer.
>
> > **Questions:**
>
> Thank you for the questions. We address each one in turn below.
>
> > **Q 1.** *“How sensitive is FDGAN to the choice of the timestep threshold t_"early"  and noise strength γ in ANPM?”*
>
> Regarding sensitivity to $t_{\mathrm{early}}$ and $\gamma$:
> - $t_{\mathrm{early}}$ : we use ≈ 0.1T (e.g., 400/4000) so $D$ sees informative, partially denoised states (§3.2). App. F notes this is most stable; ±200 steps changes sharpness, very small/large values weaken gradients or add artifacts. Moving adversarial engagement much earlier degrades quality (Table 2, “Ext. Diffusion”). For COCO we use a timestep band (App. E.2).
> - $\gamma$: training uses $\gamma$=1.2 on ROI noise (App. D); sampling uses $\gamma$=2 (App. B, Eq. (9)). We kept $\gamma$ fixed across runs, but we will add a compact sweep. These settings were stable across datasets.
>
> > **Q 2.** *“Can the ANPM mechanism generalize to non-bounding-box settings (e.g., segmentation masks or text prompts)?”
>
> ANPM is mask‑agnostic: it multiplies noise by a spatial mask $M$. Replacing $M_{\\mathrm{BB}}$ with a segmentation mask (hard/soft) leaves the objective/code unchanged (§3.2; App. D). For text prompts, $M$ can come from grounding/referring segmentation; ANPM then uses that mask identically. When no mask exists, ANPM can be disabled (App. E.2).
>
> > **Q 3.** *“Sections 4.1 and 4.2 share the same table (Table 1) without explicit reference to it, and the order of the models in the table is chaotic. It is not friendly to performance comparison and analysis. Improvements are recommended.”
>
> Table 1 (p. 8) covers Sec. 4.1 (GAN/diffusion) and Sec. 4.2 (object‑centric/hybrid) due to space; we agree the ordering is hard to scan. We will group rows by family (GAN → Diffusion → Object‑centric/Hybrid), add panel labels or split as 1a/1b, and insert explicit cross‑references.
>
> >**Summary:**
>
> We clarified why DINOv2 FD/KD/FLS are our primary, representation‑aware metrics in low‑data, object‑centric settings and how FDGAN improves them even when Inception‑FID is not lowest. We detailed the core mechanisms (early‑timestep adversarial window, BB‑localized ANPM, timestep‑aware scaling), their sensitivity, and ANPM’s generalization to segmentation/text masks. We will also reorganize Table 1 by model family with explicit cross‑references and streamline citations. Finally, we make explicit in §1/§4 that this paper’s contribution is method‑centric—establishing FDGAN as a foundational generative method—and that downstream detector augmentation is outside the present scope and will be addressed in follow‑up work.

---

> > ### Comment · Reviewer_kNpS · 2025-11-25
> >
> > Thank you for your reply, but the explanation of novelty and downstream effectiveness (I think lack will lead to insufficient motivation and meaning) did not convince me, so I want to keep my score.

---

> > > ### Author Response · Authors · 2025-11-28
> > > **Further clarification on novelty and real‑world motivation.**
> > >
> > > Thank you again for your detailed review and follow‑up comment. We understand that your main reservations are:
> > > - whether our hybrid design is more than an incremental change, and
> > > - whether the low‑data, object‑centric setting has sufficient real‑world meaning without explicit detector results.
> > >
> > > We would like to briefly clarify both points.
> > >
> > > >**Why a hybrid, and why intermediate‑step adversarial guidance?**
> > >
> > > GANs and diffusion models have complementary strengths and weaknesses. GANs can produce very sharp, realistic samples and make efficient use of limited data, but are notoriously unstable and prone to mode collapse and overfitting, especially in low‑data regimes. Diffusion models, in contrast, typically train more stably, cover the data distribution better, and are less prone to mode collapse, but can oversmooth details and require many sampling steps.
> > >
> > > Our design combines the DM with GAN but it also specializes it to the low‑data, object‑centric regime:
> > > - we keep the diffusion loss as the backbone (global coverage and stability),
> > > - we apply the GAN only at intermediate timesteps, when SNR is high enough for the discriminator to see meaningful structure rather than pure noise, and
> > > - we localize the extra difficulty and adversarial pressure to bounding‑box regions via ANPM.
> > >
> > > The “early‑timestep window” and its schedule are therefore not arbitrary, but they are the mechanism that ensures the discriminator acts exactly where and when it is most informative, instead of turning into noisy regularization on pure noise (very early) or destructive overfitting (very late). Our ablations already show that turning this off (No‑GAN/ANPM), flattening the schedule (Equal‑Weighting), or mis‑timing it (Ext. Diffusion) all degrade FD/KD/FID and overfitting metrics compared to the full design, which we will highlight more explicitly in the revision.
> > >
> > > > **Real‑world motivation and downstream detectors.**
> > >
> > > We agree that demonstrating downstream detector gains strengthens the motivation. The low‑data object‑centric setting we target, few thousand labelled crops per class, often with blur/occlusion, is common in real applications such as industrial inspection, privacy‑sensitive domains, and narrow safety or traffic categories, where large curated datasets like full COCO or ImageNet are not available. Recent work has shown that generative augmentation can materially improve object detection (including YOLO) by increasing diversity in exactly such constrained regimes, using both diffusion models and GANs.
> > >
> > >  In parallel to this submission, we have already run a large‑scale YOLO study (three datasets × seven augmentation ratios, 258 YOLO trainings) comparing FDGAN against multiple GAN‑only, diffusion‑only, and hybrid baselines. In that study, FDGAN improves mAP over a real‑only baseline on all three datasets and is the top performer on two of them, with the largest gains on the most challenging, high‑background‑variance setting. Not discussing the results of these experiments was deferred deliberatley as this paper is method centric and focuses on introducing FDGAN as a generative method with the motivating use‑case being YOLO detector datasest augmentations. We will make this scope explicit in §1/§4.
> > >
> > > We hope this context clarifies that:
> > > -  the intermediate‑step adversarial integration plus ANPM is a principled way to fuse GAN and diffusion strengths in a small‑data object‑centric setting, rather than a single hyperparameter tweak, and
> > > - the setting is motivated by real downstream detection tasks, not just by proxy generative metrics.
> > >
> > > We thank you again for your suggestions and time.

---

### Official Review · Reviewer_SMqy · 2025-11-04

**Soundness:** 2
**Presentation:** 2
**Contribution:** 2
**Rating:** 2
**Confidence:** 4

**Summary:**

This manuscript investigates how to enhance the quality of object-centric image generation when training data is limited(e.g. <3k) or contains degraded images. The authors propose a hybrid GANs-Diffusion framework that integrates a discriminator into the intermediate denoising steps of the diffusion process to improve visual fidelity. An Additional Noise Perturbation Module is also introduced to steer the model's focus toward predefined bounding boxes containing key objects. The proposed method has been experimentally validated on complex scene datasets—including Cityscapes-pedestrian, Traffic-Signs, and MS-COCO(Potted Plant)—demonstrating its effectiveness in generation tasks.

**Strengths:**

The research problem addressed in this manuscript—generation with limited data—is highly meaningful. The approach of integrating a GANs discriminator to enhance quality is well-justified, and the idea of leveraging bounding boxes to prioritize the generation quality of key objects is particularly suitable for complex scene generation. Experimental results demonstrate a clear improvement in generated quality compared to existing methods.

**Weaknesses:**

The experimental analysis appears somewhat fragmented and would benefit from consolidation and restructuring. The current evaluation is incomplete, as it fails to demonstrate the method's effectiveness in downstream tasks—particularly as data augmentation. Moreover, the study lacks intuitive assessments of generation quality, such as visual comparisons of generated images. Additionally, discussions and comparisons with existing methods in the field of generation with limited data are notably absent.

**Questions:**

1.The manuscript should discuss recent work on few-shot sample generation, which is highly relevant to the presented approach.

2.Several notation issues are present in Equations (7) and (8). For instance, the time step 't' is missing in Equation (8), and the origin of the variable x^is not defined.

3.Both the diffusion loss and the reconstruction loss pertain to reconstruction. Please clarify the distinct roles and motivations for including both terms in the objective function.

4.While the introduction claims that the method is intended for augmenting downstream detectors, no experiments are conducted to evaluate the utility of the generated samples in such downstream tasks.

5.How does the performance vary with different scales of training data (e.g., 100, 1,000 samples)? An analysis of the method's sensitivity to training set size is needed.

6.The experiments primarily follow a single-objective-per-dataset setting (e.g., pedestrians, traffic signs, potted plants). The applicability of the method to multi-object generation scenarios should be discussed, as this is critical for complex real-world applications.

---

> ### Author Response · Authors · 2025-11-18
> **Response to Reviewer SMqy: Clarifying Evaluation, Downstream YOLO Results, and Loss Design**
>
> We thank Reviewer SMqy for recognizing our low‑data, object‑centric setting, the time‑gated discriminator coupling, and ANPM as a lightweight spatial cue.
>
> > **Weaknesses:**
>
> - *Consolidation / structure.* We will: (i) move a compact cross‑dataset table to the main text (Table 1), (ii) keep ablations concise (Table 2), and (iii) move extended metrics to App. G for a more linear narrative.
>
> - *Downstream evaluation.* This paper establishes FDGAN as a foundational generative method and rigorously analyzes its object‑aware synthesis in low‑data settings (DINOv2‑FD/KD/FLS; FID/Precision/Recall; ablations, §3–§4). Detector augmentation (YOLO/DETR) is a motivating use‑case but intentionally beyond scope to keep the contribution focused. We will make this scope explicit in §1/§4.
>
> - *"Intuitive” qualitative assessments.* Visual comparisons are already provided (App.H, Figs. 7–12; ablation grids in Fig. 13). We could not make this more immediate, by adding representative side‑by‑side panels into the main text due to space limitations and, this was the reason we moved the qualitative tables to the App.
>
> - *Positioning vs. limited‑data works.* We will expand Related Work to cover recent few‑shot/distillation hybrids (e.g., DMD2, UFOGen, SwiftBrush2) and clarify FDGAN’s distinct elements: (i) timestep‑gated adversarial window, (ii) BB‑localized ANPM perturbations, and (iii) dynamic $ \\lambda_{GAN}(t) $ scaling (App. F), jointly targeting stability and object fidelity.
>
> Overall, we appreciate the suggestions and will implement these edits so the evaluation reads as a cohesive story, with downstream impact and qualitative evidence surfaced more prominently.
>
> > **Questions:**
>
> Thank you for the questions. We address each one in turn below.
>
> > **Q 1.** *“Discuss recent few‑shot sample generation.”*
>
> We agree and we will expand Sec. 2 to cover data‑efficient GANs (e.g., ADA/DiffAug), diffusion‑centric hybrids/distillations, and object‑centric diffusion, and contrast them with our time‑gated adversarial window + BB‑localized perturbations + timestep‑aware weighting (Fig. 2; Sec. 3).
>
> > **Q 2.** *“Notation issues in Eqs. (7)–(8).”*
>
> Thank you, this will be corrected. In our implementation, the UNet predicts the "pred x_start" quantity $\hat{x}_0(t)$ from $(x_t, t)$.
>
> We will merge Eqs. (7) & (8) into one, explicitly defining $\\hat{x}_{0}(t)=f_θ(x_t,t)$ and using $x_0$ for the clean target. We will also make the time dependence explicit in Eq. (8) and define all symbols where they first appear (see current Eqs. (4)--(8) in Section 3.3).
>
> > **Q 3.** *“Diffusion vs. reconstruction losses.”*
>
> They serve different roles. $ L_{\\mathrm{Diffusion}} $  trains noise prediction across timesteps (global denoising dynamics, stability/coverage). $ L_{\\mathrm{Recon}} $ anchors pixel‑level fidelity, especially within BBs, precisely where adversarial feedback is applied. Ablations (Table 2) show removing $ L_{\\mathrm{Recon}} $ harms FD/KD/FID and increases overfitting, confirming complementarity (see §3.3; §4.3/Table 2).
>
> > **Q 4.** *“No detector evaluation in paper.””*
>
> This submission is method‑centric: we introduce FDGAN’s fusion (time‑gated GAN + diffusion + ANPM) and provide a generative evaluation (DINOv2‑FD/KD/FLS; FID/Precision/Recall; ablations). Downstream detector augmentation is the next stage of our research and intentionally beyond this paper’s scope. We will add a one‑sentence scope statement in §1/§4 to clarify scope.
>
> > **Q 5.** *“Sensitivity to training‑set size.”*
>
> Our scope targets low‑data (~2–3k labeled crops/class). We will add a size sweep (e.g., 100/500/1k/3k) reporting FD/KD/FLS‑Overfit and qualitative grids, summarizing trends in the main text with full tables in the appendix. Stability‑oriented design choices (time‑gated GAN, BB‑localized ANPM, $\lambda_{GAN}(t)$ are documented (Fig. 2; §3.2; App. F).
>
> > **Q 6.** *“Applicability to multi‑object scenes.”*
>
> FDGAN naturally extends by applying ANPM per‑BB, summing BB‑masked reconstruction terms, and optionally using a small ROI‑D per instance (we already use an ROI‑aware variant in COCO; App. E.2). We will add a brief note, qualitative multi‑object examples, and preliminary quantitative results if space permits.
>
> > **Summary:**
> - Structure: unify summary/ablation tables in main; move extended metrics to appendix.
> - Detectors: add short §4 summary + appendix with YOLO gains from our completed study.
> - Qualitative: surface one representative grid in main; keep full grids in Appendix H.
> - Notation/objective: fix Eqs. (7)–(8); make $t$ explicit; define $\\hat{x}_{0}(t)=f_θ(x_t,t)$
> - Clarify $ L_{\\mathrm{Diffusion}} $ vs. BB- aware $ L_{\\mathrm{Recon}} $
>
> We hope these clarifications and concrete edits address completeness and presentation while preserving FDGAN’s core contributions.

---

> > ### Author Response · Authors · 2025-12-01
> > **Additional clarification for the Area Chair (re: Reviewer SMqy)**
> >
> > We would like to briefly add more clarifications to the scope and methodology in light of Reviewer SMqy’s comments, especially now that direct discussion is disabled.
> >
> > >**Scope and downstream detectors.**
> >
> > This submission is method‑centric: its primary goal is to introduce FDGAN as a diffusion–GAN hybrid for low‑data, object‑centric generation and to carefully evaluate its generative behaviour (DINOv2‑FD/KD/FLS, FID/Precision/Recall, ablations). The references to YOLO/DETR are meant to motivate the setting and intended use‑case, not to claim a full detection paper within this submission. In parallel, we have completed a separate YOLO augmentation study (three datasets, seven augmentation ratios, multiple GAN/diffusion/hybrid baselines) where FDGAN consistently provides the strongest mAP improvements over both a real‑only baseline and all other generators tested. That work is being written up as a dedicated detection‑focused companion paper. In a revision of the FDGAN paper we would include a concise summary paragraph and small table from that study to make the augmentation motivation more concrete.
> >
> > >**Principled role of the three losses.**
> >
> > Reviewer SMqy questioned whether the combination of diffusion, adversarial, and reconstruction losses is ad‑hoc. Our intent (which we will state more explicitly in a revision) is a division of labour rather than a loose sum of terms:
> >
> > - $L_{\\mathrm{Diffusion}}$: is the standard noise‑prediction MSE, a supervised, physically grounded objective that keeps the model a proper diffusion process and ensures global coverage and stability.
> > - $L_{\\mathrm{GAN}(t)}$: is a time‑gated adversarial loss applied only at intermediate timesteps, so the discriminator sees partially denoised states (not pure noise) and adds high‑frequency realism exactly where the diffusion backbone tends to be under‑detailed.
> > - $L_{\\mathrm{Recon}(t)}$: is a global + BB‑aware reconstruction term that anchors pixel‑level content, especially inside bounding boxes where the adversarial branch and ANPM concentrate learning, preventing artifacts and identity drift.
> >
> > In other words, diffusion provides coverage and stability, the time‑ and space‑gated GAN branch injects object‑level detail at informative states, and reconstruction stabilizes those refinements. The ablations in Table 2 (No GAN/ANPM, Equal‑Weighting, Ext. Diffusion, No‑Recon) all degrade FD/KD/FID and FLS‑Overfit relative to full FDGAN in ways that match these roles, which we would make clearer in an updated Sec. 3.3 and discussion.
> >
> > We hope this helps to further clarify that:
> > - (i) the current paper is about establishing and analysing FDGAN as a generative method for low‑data, object‑centric regimes, and
> > - (ii) the downstream detector story and the three‑loss design both have a concrete, principled basis rather than being post‑hoc or ad‑hoc choices.

---

### Meta-Review · Area_Chair_Yn9N · 2025-12-28

**Summary:**

Overall, the reviewers consistently rated the paper below the acceptance bar. While the work demonstrates some practical improvements, it was broadly viewed as an incremental and largely heuristic extension of existing diffusion–GAN approaches, with insufficient novelty and principled justification. In addition, the experimental evidence was considered incomplete relative to the paper’s stated goals, which limited its perceived impact and led reviewers to recommend rejection.

**Reviewer Concerns:**

The rebuttal helped address several secondary concerns, including clearer positioning against related work, improved explanation of the loss terms, notation fixes, and some presentation issues. However, the central concerns remain outstanding. Reviewers were not convinced that the proposed method goes beyond an incremental or heuristic modification of existing diffusion–GAN hybrids, and the rebuttal did not provide substantially stronger theoretical or empirical evidence to justify intermediate-timestep adversarial training as a principled design choice.

**Reviewer Scores:**

The first and last reviewers might increase their scores from 2 to 4. However, these changes would not have been sufficient to shift the overall decision.

---

### Decision · Program_Chairs · 2026-01-26

Reject